# Comparison of potential drinking water source contamination across one hundred U.S. cities

Sean W. D. Turner [1✉], Jennie S. Rice [1], Kristian D. Nelson[1], Chris R. Vernon [1], Ryan McManamay [2], Kerim Dickson [3] & Landon Marston[4]

Drinking water supplies of cities are exposed to potential contamination arising from land use and other anthropogenic activities in local and distal source watersheds. Because water quality sampling surveys are often piecemeal, regionally inconsistent, and incomplete with respect to unregulated contaminants, the United States lacks a detailed comparison of potential source water contamination across all of its large cities. Here we combine national-scale geospatial datasets with hydrologic simulations to compute two metrics representing potential contamination of water supplies from point and nonpoint sources for over a hundred U.S. cities. We reveal enormous diversity in anthropogenic activities across watersheds with corresponding disparities in the potential contamination of drinking water supplies to cities. Approximately 5% of large cities rely on water that is composed primarily of runoff from non-pristine lands (e.g., agriculture, residential, industrial), while four-fifths of all large cities that withdraw surface water are exposed to treated wastewater in their supplies.

[1] Pacific Northwest National Laboratory, Richland, WA, USA. [2] Baylor University, Waco, TX, USA. [3] Kansas State University, Manhattan, KS, USA. [4] Virginia Polytechnic Institute and State University, Blacksburg, VA, USA. ✉email: sean.turner@pnnl.gov

As the demand for clean, piped water grew rapidly in the United States during the late 19th and early 20th centuries, dozens of major cities constructed centralized systems of surface water supply that often featured inter-basin transfers and reservoirs to import and store water from remote river basins[1]. These storage and conveyance systems were built during an era when the contributing source watersheds were mostly free from activities that could contaminate surface water. Today, however, source water contamination is widespread. Almost none of the nation's surface water is drinkable without treatment[2] and health-based water quality violations affect up to 45 million people annually[3]. Nitrates from agricultural runoff contaminate the water supplied to millions of U.S. residents[4] and the number of water supply systems experiencing nitrate concentration violations is increasing[5]. Poly- and perfluoroalkyl substances (PFAS)—which leach from airports, oil refineries, military installations, and manufacturing facilities—are present at above-advisory levels in the drinking water of ~6 million U.S. residents[6] and are difficult to remove during treatment[7]. Treated wastewater outflows are present in half of the nation's drinking water treatment plant intakes[8], carrying the risk of exposing consumers to *Cryptosporidium* and norovirus infection[9] as well as pharmaceuticals and endocrine disrupting compounds (EDCs)[10]. Importantly, the teleconnections between water quality at supply system intakes and the anthropogenic activities occurring in source watersheds create significant challenges for urban water utilities. Many cities are forced to invest in advanced treatment technologies or seek alternative water supplies that increase water prices[11].

Despite the importance of source water protection to utilities and their customers, the data available for performing a national comparison of potential source water contamination across cities present a number of limitations. The Clean Water Act (CWA) Section 303(d) list of impaired water bodies provides information on whether surface waters meet a specified standard for a range of purposes, including drinking water. However, these determinations are made by individual states, each using its own methodology[12]—making the data unsuitable for national-scale comparison. Regulations require that each major water utility in the U.S. report water source details and post-treatment water quality parameters, yet consumers lack information on the characteristics, prevalence, and impacts of human activities occurring in lands upstream of reservoirs and river intakes. These details are important because national water quality sampling surveys, such as National Aquatic Resource Assessments, focus on regulated or well-established contaminants. Sampling surveys thus tend to overlook unregulated or currently unrecognized water quality impairments[13]. By instead studying the presence of human activities in source watersheds, one can account implicitly for the gamut of potential contaminant sources that could affect water quality. Analysis of anthropogenic activity in source watersheds could inform water utility, state, and federal drinking water protection efforts and be used in conjunction with geospatial projections of land use and land cover change to identify new or increased future pressures on water quality.

Our study offers an approach for analyzing and comparing individual U.S cities' exposures to potential drinking water supply contamination from multiple point and nonpoint sources. We employ a high-resolution, geospatial analysis of multisectoral land use data with national coverage, combined with simulated 1/24° grid resolution runoff and regulated flows at water supply intakes (see "Methods"). Our analysis covers the 116 largest U.S. cities, each with over 150,000 inhabitants (Supplementary Fig. 1 lists all studied cities). These cities constitute a quarter of the U.S. population[14], and, in contrast to less populous areas, often

withdraw and blend water from intakes located across multiple local and remote watersheds and aquifers[15].

To quantify potential point and nonpoint anthropogenic contamination for these cities, we first combine spatially referenced regions of drinking water supply catchments with geospatial layers detailing the presence of human activities—namely croplands, economic sectors, industrial facilities, human settlements, and wastewater treatment plants. In contrast to previous research focused on individual water treatment plant intake locations[16,17], we evaluate the potential contamination of each city's total water supply, accounting for the relative contributions of water supply sources from multiple watersheds. For instance, a water body subjected to cropland runoff would imply a greater percentage of potential contamination if it served as a city's entire water supply than if it contributed only a small proportion of the overall water supply. To capture this dynamic, we enhance a geodatabase of urban water supply catchment delineations[18] with estimates of their relative average contribution to public supply for each city. We obtain these source water contribution estimates, which include groundwater, for all 116 cities via publicly available utility documents and websites (see "Data availability"). These data highlight the prominent role played by surface water resources in supplying large U.S. cities. The average large city relies on surface water for 81% of its supply. Three quarters of large cities rely more on surface water than groundwater (including nine of the 10 largest U.S. cities by population) and surface water makes up 100% of supply for more than half of large cities (Fig. 1).

These city-scale source water contribution breakdowns allow us to create two metrics that characterize the potential contamination in a city's overall water supply. These are: (1) the Nonpoint Proportion of Potentially Contaminated Supply (Nonpoint PPCS, %), defined as the percentage of city water supply generated as runoff from non-pristine lands—meaning land developed for either agricultural purposes or non-agricultural human uses (urban, residential, infrastructure, industry), and (2) the Point PPCS (%), defined as the percentage of city water supply comprised of effluent from municipal wastewater treatment plants. For a given watershed, we define the Point PPCS as the aggregated average discharge from all wastewater treatment plants as a fraction of the average regulated flow at the intake or reservoir serving the public water supply system. We estimate both Nonpoint and Point PPCS at the scale of an individual city's supply system by accounting for the relative contributions of each water resource, including groundwater, to each city's total drinking water supply. We do not address potential groundwater contamination in this study (see "Methods" for implications). Since we lack volumetric discharge data for point sources other than municipal wastewater treatment plants (e.g., mines), the Point PPCS results in our study equate to "de facto wastewater reuse" (see Table 1 for definitions of wastewater reuse types). Previous literature recommended an assessment of de facto reuse at supply system level[19], but, until our research, this depth of analysis had yet to be developed for a national scale study.

In addition to our Point and Nonpoint PPCS metrics, we analyze the presence and density of multisectoral land use activities in all watershed areas upstream of the key intakes supplying each city. This includes land areas used for both agricultural and non-agricultural purposes (e.g., urban areas) as well as the density of point polluter facilities (e.g., mines) registered with the EPA's National Pollutant Discharge Elimination System (NPDES) and Toxic Release Inventory (TRI) permit programs. To our knowledge, this study provides the most comprehensive set of rankings to date for large U.S. cities according to their

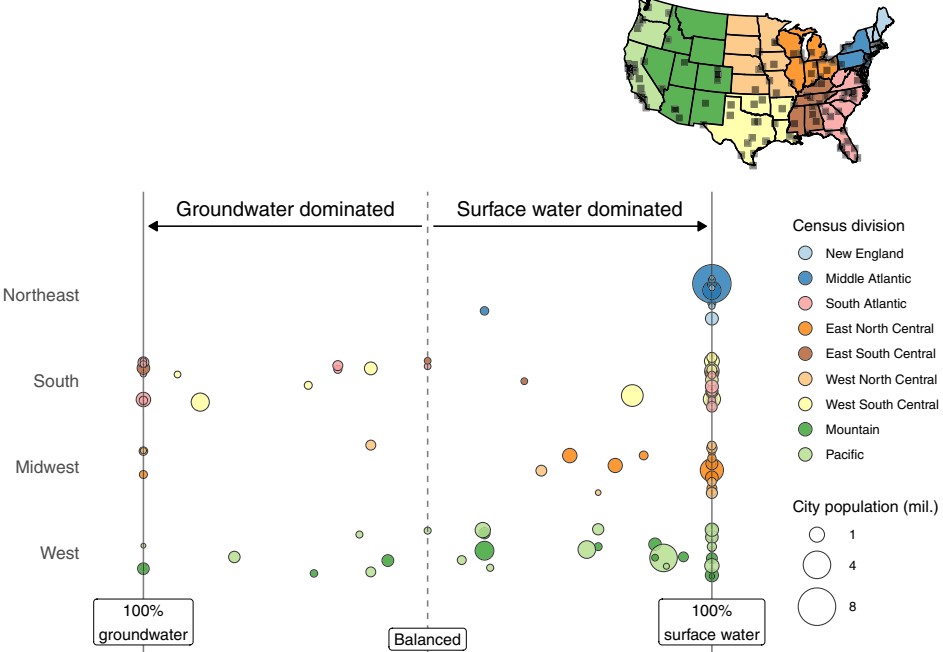

**Fig. 1 Surface water dominates urban water supply.** Relative contributions of surface water and groundwater to total water supply across 116 United States cities with population exceeding 150,000 people (city locations shown on map, top right). Vertical positioning of each point within each Census region (West/South/Northwest/Midwest) is random to avoid overlap.

**Table 1 Definitions of planned and unplanned forms of wastewater reuse.**

| Wastewater reuse type | Planned/unplanned | Mechanism |
|---|---|---|
| De facto reuse | Unplanned | Wastewater treatment plant effluent is discharged to surface water and later drawn into a downstream city's water treatment facility. |
| Indirect potable reuse (IPR) | Planned | Wastewater treatment plant effluent is purified using advanced water treatment technology and then pumped back into the existing water supply source (same city), where it mixes with in-situ water before being drawn back into supply via existing infrastructure. |
| Direct potable reuse (DPR) | Planned | Wastewater treatment plant effluent is purified using advanced water treatment technology then supplied directly back into the distribution system serving the same city. |

exposures to different sources of potential water contamination. Datasets applied in our analysis are the most up-to-date available and sometimes represent different years within the last decade (e.g., 2016 for cropland cover, 2012 wastewater treatment plant mean discharge). Since the variables applied in our study change gradually rather than abruptly, we would expect no significant change in results if a consistent year of analysis were available.

### Results and discussion
Source watersheds serving large U.S. cities reflect the nation's wide diversity in land use and anthropogenic activities (Fig. 2). Watersheds range from pristine, with minimal human influence (e.g., Boise, New York City), to heavily exposed, with significant presence of industry, agriculture, or residential developments (e.g., Lexington, Indianapolis, Houston). Our results explore the potential water quality implications of this diversity for the 116 largest cities in the U.S for four dimensions: comparing Point and Nonpoint PPCS across all large cities (see section "Wide disparity in point and nonpoint potential contamination"), determining hotspots of land use and anthropogenic activities across source watersheds serving large cities (see section "Hotspot regions of land use pressure in water supply catchments"), examining the impact of source watershed diversity and hydrology on the PPCS results (see section "Source diversity and hydrology matter"), and analyzing how the adoption of indirect potable reuse could affect

the PPCS results in a small selection of cities (section "Evaluation of potential contamination with alternative water supply strategies: example of indirect potable reuse (IPR)"). We conclude with a discussion of the potential value of our exposure-based metrics for predicting actual contamination (see section "Potential to infer actual source drinking water contamination risk from PPCS metrics").

**Wide disparity in point and nonpoint potential contamination.** Our results highlight enormous disparity in the potential contamination of surface water supplied to U.S. cities (Fig. 3). Nonpoint PPCS ranges from 0% to ~97% (with Des Moines, Iowa the most extreme case), while Point PPCS (i.e., de facto wastewater reuse) ranges from 0% to almost 15% (Houston, Texas). Very few cities avoid a background level of Point PPCS. We find that four out of every five large cities that rely on surface water for drinking water supply experience de facto reuse (i.e., 84 out of the 104 cities with surface water).

The distributions of both Nonpoint and Point PPCS are heavily skewed toward lower proportions. In other words, most cities benefit from watershed lands that are either unsuited to significant development or have been protected by the receiving city. More than half of cities supply water generated from relatively pristine land (i.e., Nonpoint PPCS < 5%) and with less than 2% treated wastewater effluent (i.e., Point PPCS < 2%). Cities

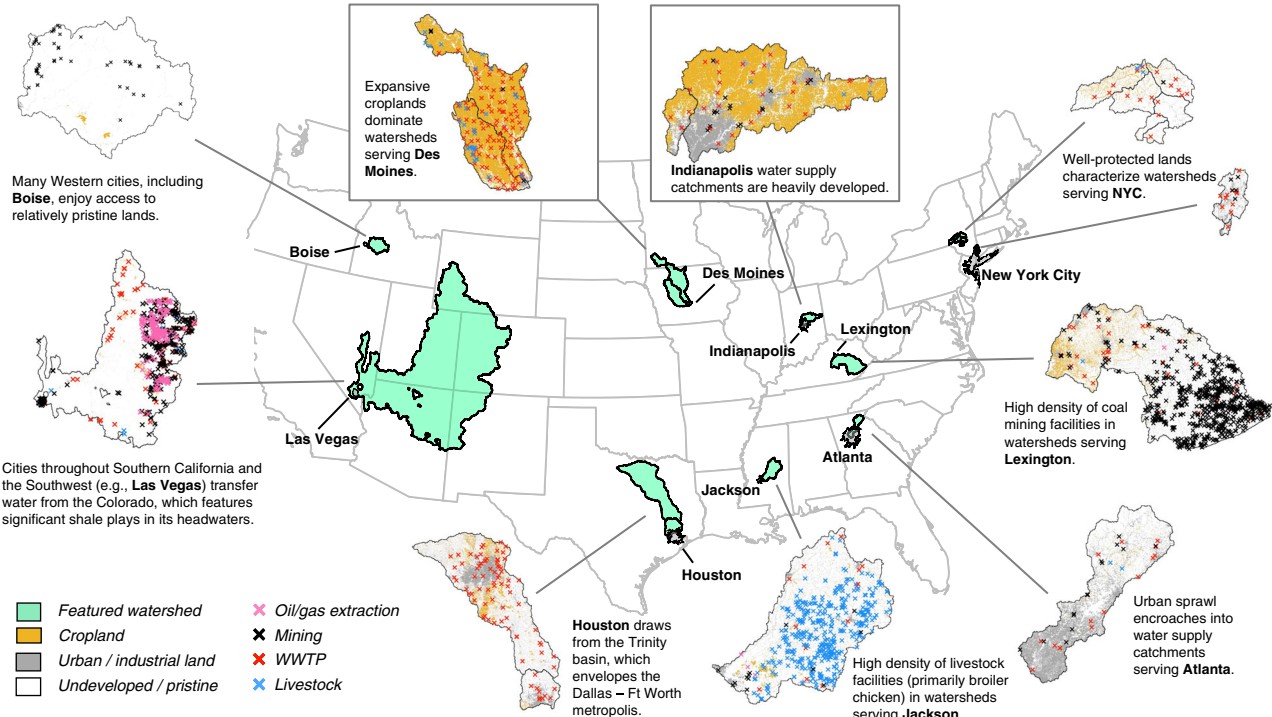

**Fig. 2 Diverse land use and anthropogenic activities across urban source watersheds.** Select cities and their associated source watersheds, showing the diversity of primary land use and anthropogenic drivers of potential contamination across the U.S.

exposed to relatively high levels of potential contamination from either point or nonpoint sources tend to be located throughout the Midwest, the South, and Texas. Although Point and Nonpoint PPCS are somewhat correlated (Spearman's correlation ~0.5), very few cities occupy the extreme high side of both metrics. The cities of Indianapolis (Indiana) and Atlanta (Georgia) are the only two cities ranked in the top ten for both Point and Nonpoint PPCS.

Twelve cities have Nonpoint PPCS > 20% and are all located in either the Midwest (seven cities) or South Atlantic regions (five cities). Cities in the top quartile (i.e., >10% Nonpoint PPCS) that lie outside of these two regions tend to rely on watersheds that drain large areas of the Midwest. These include New Orleans (Louisiana), drawing water at the mouth of the Mississippi River, and Buffalo (New York), drawing from Lake Erie. The Midwest region dominates the extreme end of Nonpoint PPCS, due to its expansive croplands. While median Nonpoint PPCS across all 116 cities is ~1.7%, four Midwestern cities—Des Moines, Indianapolis, Fort Wayne (Indiana), and Columbus (Ohio)— have >60% Nonpoint PPCS (Fig. 3b). High Nonpoint PPCS cases in the South Atlantic region arise due to significant urbanization of the watershed—common when drinking water intakes are within city boundaries.

Like Nonpoint PPCS, the distribution of Point PPCS is heavily skewed, with 38 of 116 cities <0.1% and only six >5% (median Point PPCS ~0.5%) (Fig. 3c). The highest scoring cities for Point PPCS are Houston, Dallas (Texas), Salt Lake City (Utah), Macon (Georgia), Toledo (Ohio), Cleveland (Ohio), and Buffalo. The latter three rely entirely on Lake Erie for public water supply, with equal Point PPCSs of 5.5%. Houston's Point PPCS is 14.4%—by far the largest of all cities examined. Houston's primary supply source is Lake Livingston in the Trinity River basin, which captures wastewater treatment plant effluent from Dallas and Fort Worth (Texas) upstream. Dallas has the second highest Point PPCS, due to significant residential sprawl and the associated wastewater treatment facilities discharging within Dallas's water

supply catchments. This urban sprawl means Dallas also appears in the top quartile for Nonpoint PPCS.

Cities with very low Point and Nonpoint PPCS are primarily located in the West. Of the 38 cities with <1% Nonpoint PPCS (excluding the 12 groundwater only cases), 26 are in the West (distributed across the Pacific and Mountain Census divisions) and six are in western and southern Texas. Most of the 20 surface-water dependent cities with Point PPCS values of 0% (meaning no wastewater treatment discharges upstream of active intakes) are in the western and northeastern U.S. These include the major cities of Seattle (Washington), San Francisco (California), Portland (Oregon), and Boston (Massachusetts). Western cities in particular benefit from proximal mountain ranges (Sierra Nevada, Cascades, Rockies) that provide excellent dam sites, reliable flow augmentation from snowmelt, and steep terrain generally unsuitable for contamination-causing land development.

Some cities in western and southern Texas, such as Brownsville and Laredo, have very low Nonpoint PPCS despite being among the top quartile cities for Point PPCS. Large Point PPCS coincident with low Nonpoint PPCS is often associated with arid conditions. These conditions lead to a lack of natural diluting river flow, creating a relatively high proportion of effluent at intakes despite limited human presence in a watershed. Our analysis does not address lack of diluting water during drought, though this has been studied in the context of de facto reuse[17]. We can speculate as to how drought might affect potential contamination in different cases. The effects of drought on PPCS would be ameliorated to a degree in cities drawing water from storage reservoirs (or natural lakes) or with source diversity (conjunctive surface and groundwater use, or access to multiple, distal watersheds). Cities most vulnerable to low flow events and resulting spikes in PPCS are those relying on direct river abstraction with few alternative supply options. Some of the high Nonpoint PPCS cities of the Midwest lie in this category, including Des Moines (reliant on the Des Moines and Racoon

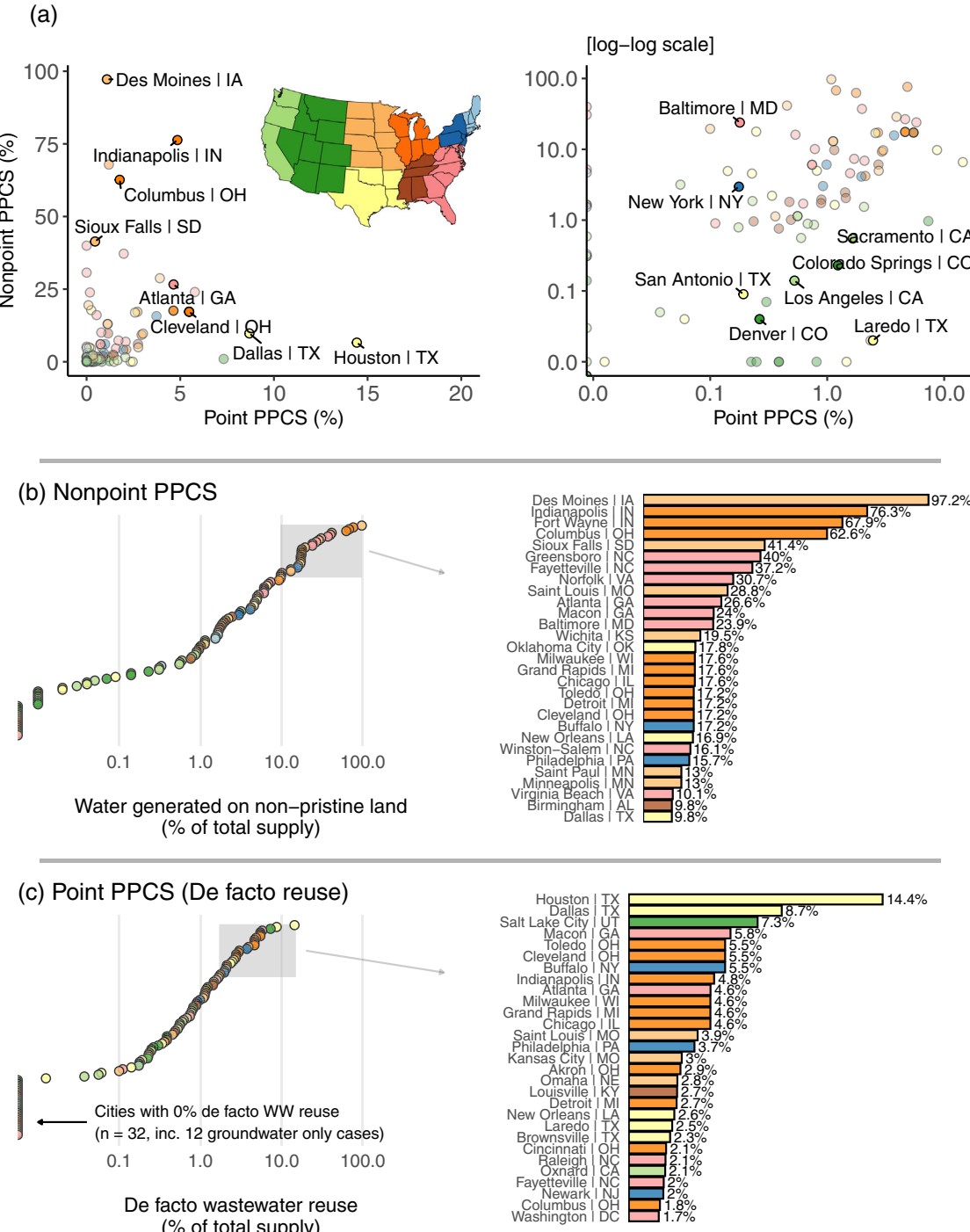

**Fig. 3 The wide disparity in potential contamination across U.S. cities.** Total water supply contamination levels for each city given in scatter plot (**a**). **b**, **c** Show the ranked distributions of nonpoint and point PPCS, respectively. In **b**, **c** the top quartile ($n = 29$) of cities is highlighted. Point color gives U.S. Census division—see map insert in (**a**).

Rivers), Fort Wayne (reliant on Saint Joseph River), and Saint Louis (Missouri) (reliant on the Mississippi River).

**Hotspot regions of land use pressure in water supply catchments.** In addition to the PPCS metrics reported above, we analyze various indicators describing the presence and density of anthropogenic activity in source watersheds. These indicators reveal hotspot regions of land use pressure, with each indicator associated with a different hotspot region (Fig. 4). Like the PPCS

results, these results are distributed with significant positive skew, featuring only a small number of cities in the upper tail. Perhaps unsurprisingly, the Midwest region dominates the extremes in cropland cover as a percentage of total water supply catchment land. Seven Midwestern cities rely on watersheds with over a third of land used for crop production (median cropland cover across all cities is ~5%). Of the 21 cities with >25% cropland cover in water supply catchments, 16 are in the Midwest and three others rely on rivers that originate in the Midwest. Interestingly, some cities located near other major crop growing regions, such

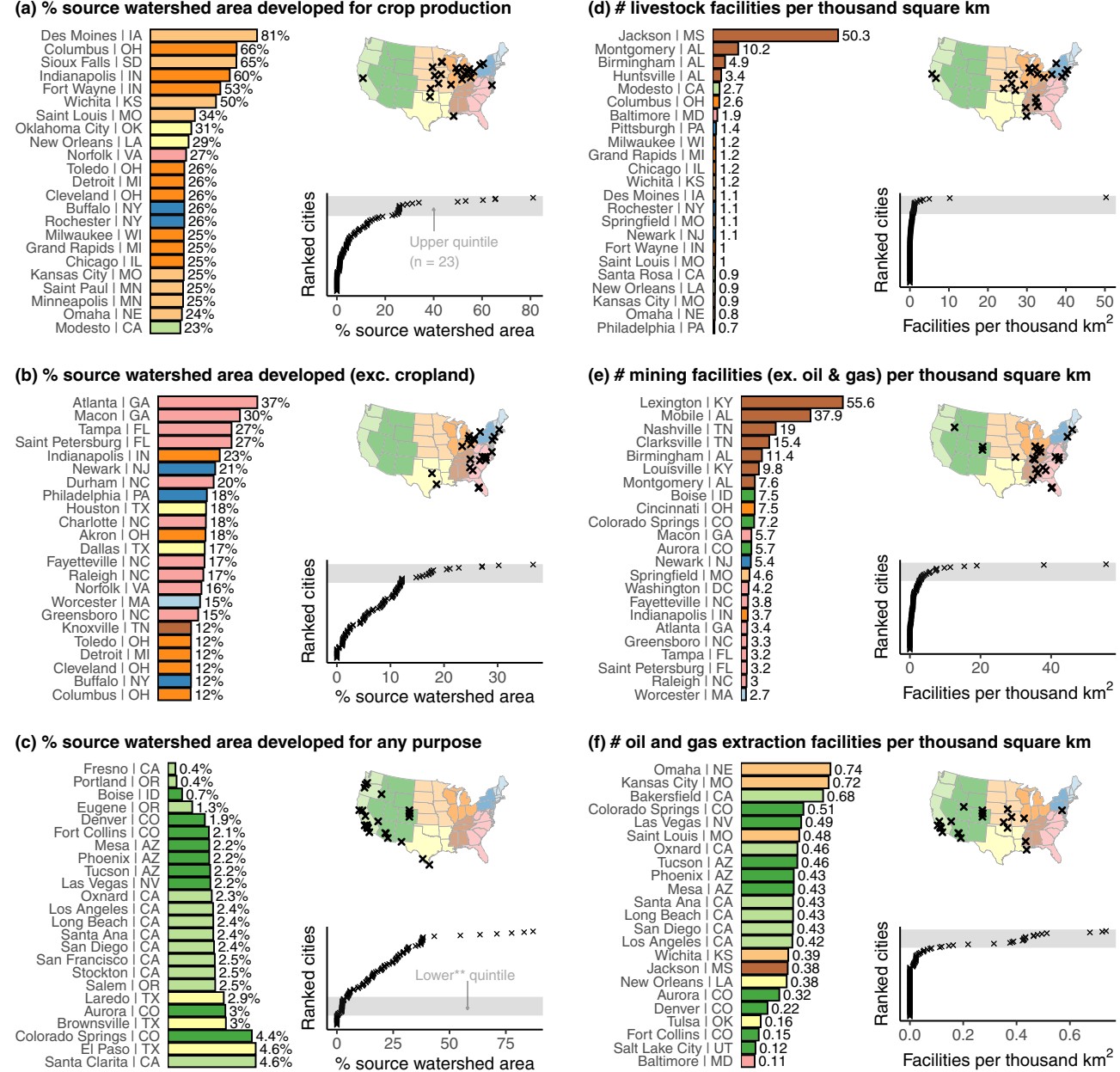

**Fig. 4 Regional hotspots of land use pressure.** Presence of land development by % watershed area upstream of city water supply intakes (**a–c**) and number of facilities registered with National Pollutant Discharge Elimination System of the U.S. Environmental Protection Agency in those watershed areas (**e**, **f**). In each panel, the cities highlighted in the map and bar chart correspond to cities of the quantile shaded in the distribution plots. This is the upper quintile (80th percentile and above) for all panels except (**c**), which highlights the lower quantile. \*\*Lower quantile excludes groundwater-only cities ($n = 12$). Bar color is by U.S. Census division (illustrated in each map).

as those in California, benefit from some of the nation's most pristine water supply catchments (Fig. 4c). Although previous research has shown that Californian public water systems experience a very high rate of drinking water nitrate violations in general[5,20], our results explain why some Californian cities, such as San Diego, Los Angeles, Oakland, and San Francisco (no nitrate violations on record), largely avoid these problems. These cities benefit from long-distance water transfer schemes that bring in water from less developed lands in California and the Colorado River Basin. In contrast, Californian cities located in the Central Valley that rely on local groundwater, such as Modesto and Fresno, are among the few cities in our analysis with recorded nitrate violations (see Supplementary Information Table 1).

While agricultural land use can dominate water supply catchments, non-agricultural anthropogenic land use (e.g., urban, industry, road networks) is rarely as pervasive. For example, no city has >50% non-agricultural anthropogenic land use in its source watersheds (Fig. 4b). Nonetheless, most cities' water supply catchments contain at least some level of non-agricultural development, like roads. Median watershed land dedicated to non-agricultural development is ~8%. Seven cities rely on watersheds with more than 20% non-agricultural land development; five are in the South and Middle Atlantic regions. These two regions also account for 13 of the 17 cities with over one eighth of non-agricultural development in watersheds. Two notable exceptions from outside this region with high non-

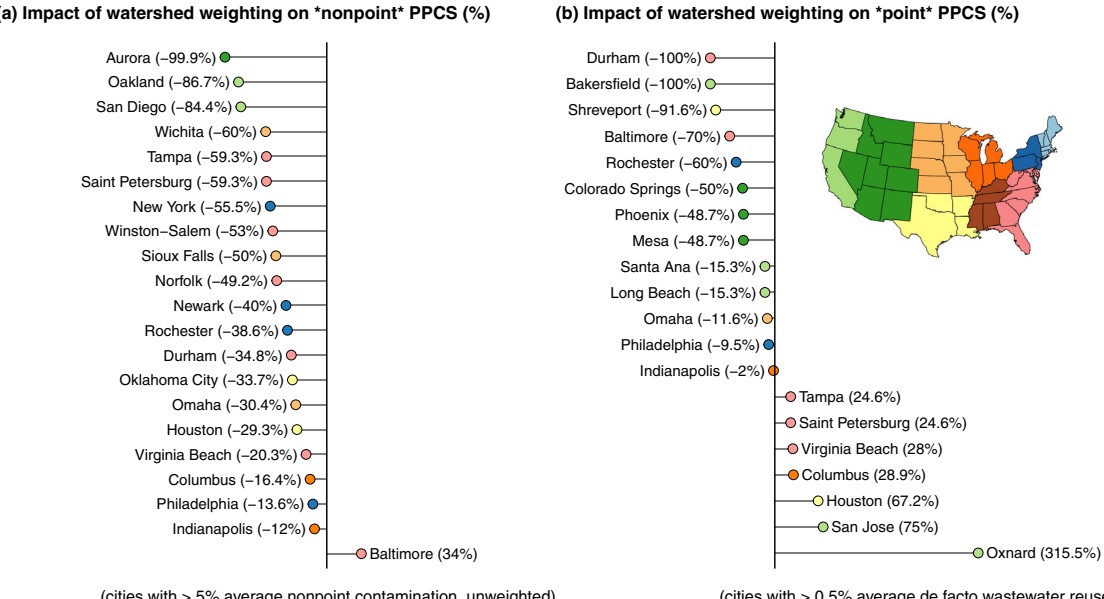

**Fig. 5 Source diversity and reliance matters. a** Displays the impact of watershed weighting on nonpoint PPCS, while **b** displays the impact of watershed weighting on point PPCS. These results show the % impact on Point and Nonpoint PPCS relative to the same metrics but calculated assuming equal weighting of supply across sources. Results are shown for cities with >5% average nonpoint contamination and cities with >0.5% average point contamination across contributing watersheds. This analysis excludes the influence of groundwater. Point color gives U.S. Census division (see map insert).

agricultural development percentages in their watersheds are Houston and Dallas (highlighted above as the cities with the largest Point PPCS). Indianapolis is a rare example of a city that relies on watersheds with high levels of both agricultural and non-agricultural land development (see Fig. 2).

Reflecting the positive skew of Nonpoint PPCS, we find that more than half of large U.S. cities benefit from water supplies generated on lands that are at least 80% pristine—meaning undeveloped for human activity. A third of large cities rely on surface water from lands that are more than 90% pristine. Large cities that draw water from the most pristine source watersheds are located predominantly throughout the Pacific and Mountain regions (Fig. 4c, which shows the lower quintile of land use development scores). Boise, a western city, benefits from nearby pristine lands for its water supply (Fig. 2). Western Texas cities in regions too arid for crop production (like Laredo and El Paso) also draw water from relatively undeveloped lands and aquifers. Rapid population growth and sprawl in many Western cities could increase pressure on these lands, so continued source protection and management efforts or development restrictions may be necessary to avoid future water quality problems.

Facility density scores—also following exponential distributions—reveal an entirely different set of regions exposed to potential water supply contamination (Fig. 4d–f). Livestock prevalence (measured by number of livestock facilities per thousand square km watershed land, with a median value ~0.1) is highest for the watersheds serving the cities in the southern band of the East South Central region, particularly in the states of Alabama and Mississippi. Specifically, the watersheds supplying Jackson (Mississippi) and to a lesser extent Montgomery, Birmingham, and Huntsville (all Alabama) feature relatively large numbers of broiler chicken facilities (Alabama and Mississippi are among the top five broiler-producing states). Jackson (Mississippi)—the most extreme case by far—has a livestock prevalence of ~50 facilities per thousand square km in its water supply catchment. The East South Central region houses the cities with the largest number of mining facilities in their water supply catchments (excluding oil and gas extraction),

specifically cities in Kentucky (Lexington, Louisville), Tennessee (Nashville, Clarksville), and Alabama (Birmingham, Mobile). The city of Lexington draws water from the Kentucky River downstream of significant coal mining facilities in the Eastern Kentucky coal fields. Similarly, Nashville and Clarksville both draw on the Cumberland River, which drains the coal fields of the Cumberland Plateau/Cumberland Mountains. Cities with high prevalence of oil and gas extraction facilities in their source watersheds are more widely distributed across the country, although the extreme cases lie in the West North Central, the Front Range, and Southern California regions. The cities of Omaha (Nebraska) and Kansas City (Missouri) rely on the Missouri River as their main source of water. The Missouri headwater tributaries contain the Bakken play of North Dakota—one of the nation's largest shale production regions, where facilities discharge treated shale gas waste into local streams. Various cities throughout the West (particularly in Arizona, Nevada, and Southern California) draw on water from Lake Havasu and Lake Mead in the middle reaches of the Colorado River basin. The Colorado headwaters hold the significant shale plays of the Piceance, Paradox, and San Juan reserves (Fig. 2).

**Source diversity and hydrology matter.** Our data show that a city's reliance on multiple source watersheds has a profound effect on potential contamination. For example, even though some cities depend on a heavily developed watershed for a portion of their water supply, access to more pristine watersheds can limit contamination exposures. Since potential contaminant levels can vary drastically across the different supply sources serving a given city, the relative weighting of these sources in total supply, herein termed "watershed weighting," affects both Nonpoint and Point PPCS (Fig. 5). For example, Rochester (New York) and Durham (North Carolina) have relatively high percentages of watershed land area dominated by agricultural and non-agricultural development, respectively, but neither city registers in the top quartile of Nonpoint PPCS. Even though Rochester draws some water from Lake Ontario—draining the vast, developed lands around the Great Lakes—the city's primary water

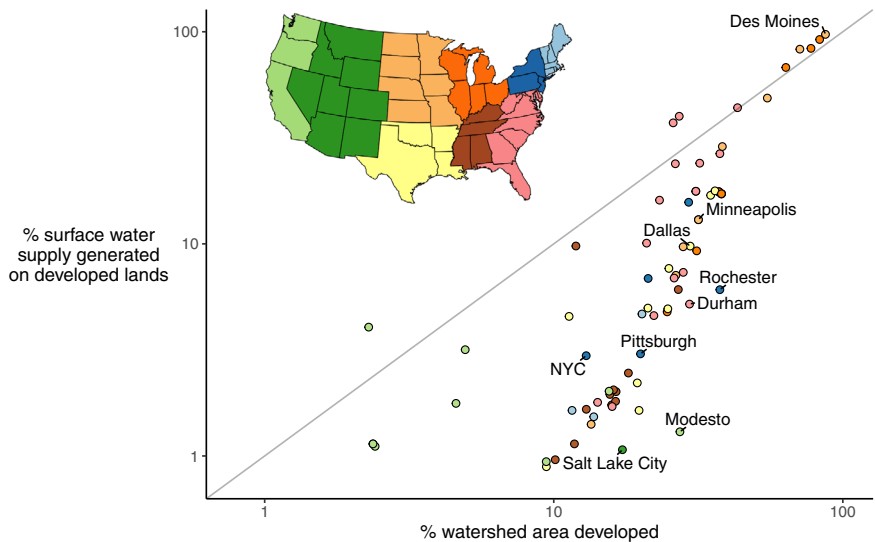

**Fig. 6 Heterogeneous watershed hydrology matters.** Comparison of % watershed area developed (agricultural and non-agricultural uses) against % of surface water supply generated as runoff on those developed lands. Point color gives U.S. Census division (see map insert).

sources are Lake Hemlock and Lake Canadice, which drain pristine lands to the south. Similarly, Durham has access to Lake Jordan, which is fed partly by runoff generated in the towns of Greensboro and Burlington. But Lake Jordan acts as an emergency-only supply for Durham; under normal conditions the city draws water from Little River Reservoir and Lake Michie, both of which drain pristine lands north of the city. For these reasons we see that both cities' Nonpoint and Point PPCS (which account for relative source contributions) are significantly lower than the simple average of anthropogenic contamination potential across resources.

We also observe that heterogeneous watershed hydrology tends to moderate Nonpoint PPCS, because both precipitation and runoff ratio vary throughout a watershed depending on meteorology, land slope, land cover, evaporation rates, etc. For instance, having 20% of a watershed devoted to urban land does not mean 20% of total runoff will be generated on urban land. Wet weather is common in high-elevation headwaters, while downstream regions featuring urbanized areas close to water intakes may be drier. We account for the impacts of watershed hydrology through the spatially distributed runoff data deployed in our study. For ~90% of cities examined, the proportion of non-pristine land area in a water supply catchment land exceeds the proportion of surface runoff generated on that land—sometimes by an order of magnitude (Fig. 6). For example, single-intake cities like Minneapolis (Minnesota) or Salt Lake City (Utah) have relatively low Nonpoint PPCS despite having relatively large proportions of watershed land devoted to non-agricultural anthropogenic development (Fig. 2).

**Evaluation of potential contamination with alternative water supply strategies: example of indirect potable reuse (IPR).** Our approach may be used to evaluate Point and Nonpoint PPCS metrics for different configurations of a supply system, such as alternative water intake locations or water supply technologies. Here we analyze the impact of hypothetical IPR implementation on PPCS for a selection of cities. We select three cities reliant on Colorado River water (San Diego, Denver, Phoenix) and three rapidly growing major cities in Texas (Austin, Fort Worth, Houston). We deem these cities likely candidates for IPR given their water supply challenges. For this exercise, we assume IPR

implementation involves advanced water treatment of a city's wastewater (becoming "reclaimed water") and subsequent return of reclaimed water to the nearest existing water supply reservoir or river intake (herein termed the "augmented source") (see "Methods"). Due to the high purity of reclaimed water, one may wish to exclude this water from the Point PPCS calculation. On the other hand, one may also wish to understand how PPCS is affected if all forms of treated wastewater (de facto and reclaimed) are included in Point PPCS. We, therefore, evaluate two versions of Point PPCS to accommodate either case. This analysis is not intended to inform costs and benefits of IPR implementation or advocate for such schemes but to demonstrate how our approach and PPCS metrics can help evaluate water supply options.

In our example, we find that meeting one third of urban water demand with IPR can have wide ranging effects on PPCS metrics (Table 2). Augmented sources with low natural inflow relative to urban water demand become predominantly reclaimed water, resulting in a significant proportion of reclaimed water in supply (~18–33% for San Diego, Phoenix, and Denver). This is in contrast to the three Texas cities, where more plentiful water at the augmented source significantly dilutes reclaimed water before it returns to the supply (~3–9% reclaimed water in supply). Utilities implementing or considering potable reuse often justify IPR (rather than direct potable reuse) by emphasizing the importance of environmental dilution in making reclaimed water acceptable to the public. Our results show that such effects can vary widely across systems.

Interestingly, an IPR scheme can reduce Point or Nonpoint PPCS in a city's overall supply. For Houston, the hypothetical IPR scheme reduces reliance on Lake Livingston, which has a very high proportion of de facto reuse. IPR in Houston reduces supply system de facto reuse and thus Point PPCS from 14.4 to 10% (assuming Point PPCS is calculated without reclaimed water). But while the IPR scheme reduces exposure to potential contamination from more distal resources, it increases exposure to potential contamination occurring at the local, augmented source. Nonpoint PPCS in Houston increases from 6.6 to 7.7%, since IPR implementation occurs at an alternative source with significant urban encroachment in the watershed.

Another interesting result for Houston is that IPR results in approximately the same proportion of total treated wastewater in supply as at present (14.7% Point PPCS if reclaimed water is

**Table 2 Hypothetical IPR (meeting one third supply) for six selected cities and its impacts on proportion of reclaimed water in supply (%) (after dilution with natural flow at the augmented source) and PPCS metrics.**

| City | Reclaimed water in supply after dilution at augmented source (%) | Point PPCS (%) | | | Nonpoint PPCS (%) | |
|---|---|---|---|---|---|---|
| | | Existing system | With IPR (excluding reclaimed water) | With IPR (including reclaimed water) | Existing system | With IPR |
| Houston | 4.7 | 14.4 | 10.0 | 14.7 | 6.6 | 7.7 |
| Fort Worth | 8.7 | 0.8 | 0.6 | 9.4 | 5.0 | 3.9 |
| Austin | 3.1 | 1.2 | 1.1 | 4.2 | 1.6 | 1.6 |
| Denver | 27.1 | 0.3 | 0.2 | 27.3 | <0.1 | <0.1 |
| Phoenix | 18.7 | 0.4 | 1.0 | 19.7 | <0.1 | <0.1 |
| San Diego | 32.9 | 0.8 | 0.5 | 33.4 | 0.9 | 0.6 |

included in the metric, versus 14.4% Point PPCS for the existing supply system). With the IPR scheme, however, ~5% of the wastewater in supply would be reclaimed water that has undergone advanced treatment. Deployment of IPR could be an acceptable water supply solution in this setting because it does not, on net, substantially increase the proportion of wastewater in supply. These results relate to a hypothetical IPR system applied to one specific local resource and do not necessarily correspond to actual expected IPR outcomes, which would depend on additional details (e.g., selection of augmented source, level of treatment, proportion of supply to be met with IPR). Nonetheless, the effects described above highlight a range of important dynamics, illustrating how local flows and the relative contributions of different sources developed here create a range of potential outcomes of an IPR scheme across different cities.

**Potential to infer actual source drinking water contamination risk from PPCS metrics**. We do not intend for our PPCS metrics and related results to imply actual contamination of water at downstream reservoirs or river intakes—our focus is on exposure to potential contamination and providing city-scale national characterization. Actual contamination implies that the anthropogenic activity generates contaminants, that those contaminants release into the environment, and that they connect to river channels and are not later absorbed, evaporated, or otherwise lost during their travel downstream. Both physical catchment characteristics and institutional factors, such as the quality and enforcement of environmental regulations, control the extent to which anthropogenic activity in a watershed leads to contamination downstream[21].

To explore the predictive value of our results, we analyze the correlation between Clean Water Act "E90" effluent exceedance violations for point polluters in various sectors and the related metrics computed in this study (Supplementary Fig. 2a–d). We standardize total effluent exceedance violations within water supply catchments by watershed area, leading to a metric of violation intensity. Unsurprisingly, we find positive correlations between violation intensities for various sectors and related metrics computed in this study. For instance, the percentage of watershed area devoted to non-agricultural anthropogenic development correlates strongly with violation intensity. Individual cases that deviate from these general relationships reveal possible differences in source protection capabilities. For example, Worcester (Massachusetts) and Raleigh (North Carolina) have a similar percentage of their source watersheds developed for non-agricultural human purposes, but the number of violations across these facilities is an order of magnitude larger per thousand square kilometer for Worcester (>1000 violations per thousand km$^2$) than Raleigh (<100 violations per thousand km$^2$) (Supplementary Fig. 2a). Exceedance violations can also be

filtered for wastewater effluent violations (labeled under the "sewerage systems" Standard Industry Code) and compared to Point PPCS (Supplementary Fig. 2c). While many cities with high point PPCS also feature among those with high intensity of sewerage system violations, significant deviations from the general relationship do exist. These discrepancies highlight the importance of the differences in human development activity type, quality of management practices adopted, enforcement of regulations, and so on, not captured in our PPCS metrics.

In addition to examining the relationships between our metrics and effluent exceedance violations, we analyze water quality violations as recorded in the EPA Safe Drinking Water Information System beginning in 1980. In contrast to the E90 violations discussed above, these water supply system violations relate to post-treatment water supplied to consumers. These violations indicate the quantity of regulated contaminants present at intakes and whether the receiving utility has the capacity to remove them. We analyze health-based violations in the following categories: Inorganic Chemicals, Synthetic Inorganic Chemicals, Arsenic, and Nitrates—in other words, contaminants that can arise from anthropogenic activities, including mining, agriculture, urban runoff, and wastewater discharges (a "health-based" violation indicates maximum concentration exceedance). Just nine of the 116 cities included in our study experienced a health-based water quality violation in the studied categories during the selected period. Four of these nine cities have high Point or Nonpoint PPCS: Des Moines (highest nonpoint PPCS), Fort Wayne (third highest nonpoint PPCS), Columbus (fourth highest nonpoint PPCS), and Louisville (top-quartile point PPCS). The other five cities with violations have relatively low point and nonpoint PPCS values. Yet these five are Western cities that use significant groundwater (Fresno, Modesto, and Santa Clarita, California, and Tucson and Mesa, Arizona). Since our PPCS metrics do not account for the potential contamination of groundwater, these inconsistencies in our results for these particular cities are not surprising.

PPCS metrics could serve as input for models aimed at predicting actual water quality. Such models could combine PPCS with data on physical watershed characteristics controlling contaminant mobilization and fate. Achieving accurate simulation of contamination would still be difficult, as the many contaminants are not measured and numerous physical factors affecting contaminant mobilization and delivery remain poorly understood[21]. Shale gas/hydraulic fracturing wastewater, for example, is a relatively understudied contaminant source, containing a range of organic compounds arising from both fracturing additives and in-situ geology as well as associated intermediates and by-products[22]. In lieu of such data and models, our geospatial metrics provide utilities and their consumers with information on anthropogenic activity in watersheds that may cause contamination and should be of significant public interest.

## Methods

**City water supply catchments.** Urban source water supply catchment delineations are obtained from the Urban Water Blueprint (UWB)[18]. Official intake data are also available through the United States Environmental Protection Agency's Safe Drinking Water Information System (SDWIS). Our independent analysis of these two data products (including comparison with source water descriptions from over 100 utility websites) leads us to opt for the UWB in this study, since we find these data the most accurate and comprehensive for describing local and distal intake locations serving large cities of the United States. In particular, we find that SDWIS intake data lack georeferencing between water supply systems and intake points located beyond each city's immediate periphery. The UWB links 235 U.S. cities to watershed delineations upstream of public water supply system reservoirs and river intakes. We filter these data for incorporated cities with population exceeding 150,000 inhabitants, identified using U.S. census data[14]. We choose this cut-off because smaller cities do not report the documentation needed to support quantified resource contributions. We rely on grey literature—primarily publicly available utility documents—to describe each city's water supply source contribution breakdown, adding our estimates of source contribution (%) in a dataset that also includes distributed runoff raster files and flow time series associated with each intake (USWCatch—see "Data availability"). For each city, all source contributions add to 100%, including groundwater and nonconventional supply sources (e.g., desalination). Existing reclaimed water schemes are not included in these contributions, since such schemes tend to be either very small scale (usually experimental projects) or are providing water for non-potable use only (e.g., landscape irrigation).

We find UWB groundwater contribution estimates inaccurate (e.g., an arbitrarily assumed 50:50 surface groundwater split is reported for many cases), so these are replaced here with estimates collected through our grey literature search. All websites and grey literature sources are listed in a supporting file included in our input data. Resources described as emergency or contingency are assumed to supply negligible water to the public supply system on average (i.e., contribution = 0%).

With each city's watershed delineations defined, we mask a range of geospatial data products to each, allowing for merging of various geospatial products to determine point effluent flows, facility density, and land areas dedicated to different forms of human development.

**Geospatial datasets deployed to assess human activity in watersheds.** Croplands generate pollution from fertilizers, pesticides, and heavy metals, which are carried into streams via surface runoff and through erosion of contaminated sediments. In the United States, agriculture is the leading cause of pollution to rivers and streams, causing high levels of nutrients to enter almost half of all streams[23]. Urbanized lands generate a variety of pollutants, including excess pesticides and nutrients from residential areas as well as oil, grease, heavy metals, salts, and toxic chemicals from motor vehicles and road runoff. To represent cropland cover, we adopt the Cropland Data Layer (CDL) 30 m raster data[24]. To represent land development (excluding agriculture) we use the National Land Cover Database (NLCD). Lands developed at low, medium, and high intensity are all included in our estimation of land area assigned for non-agricultural development (low intensity is intended to represent areas with a mixture of constructed materials and vegetation, with 20–49% impervious surfaces, including single-family housing units). Human population residing in each watershed is determined using a U.S. block population density raster layer derived from 2010 U.S. Census data[25] and is used to estimate municipal effluent for some cities.

United States polluting facilities data are obtained from the EPA Facilities Registry Service[26]. The data are filtered for facilities registered in National Pollutant Discharge Elimination System (NPDES) and Toxic Release Inventory (TRI) permit programs. Each facility has a Standard Industry Code (SIC), which allows for the identification of specific sectors present in target watersheds. We target three sectors: livestock, mining, and oil/gas extraction. Livestock operations can cause contamination from nutrients, pathogens, hormones, and antibiotics. Mining facilities—particularly abandoned coal mines—can cause acid mine drainage and metals pollution. Shale gas extraction facilities have been shown to contaminate surface water, causing high levels of chloride and suspended solids downstream[27]. The thermoelectric sector is the largest user of water by withdrawal in the United States. This sector is excluded from our analysis due to relative unimportance of thermal pollution for public water supply purposes (density of thermal plants, as well as estimated water consumption and discharges from all plants, is available in the software underlying this analysis).

Wastewater treatment plant effluent discharge estimates come from the Clean Watershed Needs Survey[28]. We use average annual municipal and industrial effluent flow estimates at the plant level. Many plants lack an average municipal effluent but report the population served by the plant. We extrapolate municipal wastewater effluent flow estimates for these plants using a linear model that predicts municipal wastewater discharge as a function of population (we fitted this model using 19,727 plants that contain both data points). Lastly, the data are filtered for wastewater plants discharging to surface waters, giving 11,917 wastewater treatment plants with effluent discharge estimates for point locations throughout the U.S.

**Runoff contributions from cropland and non-agricultural developed land.** To calculate runoff volumes generated on pristine and non-pristine areas of each city's water supply catchments, we combine the spatial layers detailing the presence of developed land (see "Geospatial datasets deployed to assess human activity in watersheds") with a CONUS-scale daily runoff dataset at 1/24° (~4 km) grid resolution forced with reanalysis climate (1980–2012)[29,30]. We aggregate in the temporal dimension to obtain a long-term average runoff rate (billion m³ per year) for each grid cell. These estimates then provide a runoff estimate for each cell of the land use raster. This allows for quantification of both total watershed runoff and total watershed runoff generated on non-pristine land. We use these data to determine the percentage of runoff generated on non-pristine land for each watershed. Employing spatially distributed runoff allows us to account for heterogeneous precipitation and water yield distribution throughout a watershed, thus capturing, for example, variable rates of runoff in a large watershed. For each city, non-pristine runoff values (%) for each watershed are used to compute a city-level Nonpoint PPCS (%). We calculate Nonpoint PPCS by taking a weighted average across watersheds that accounts for the contribution of each watershed to overall supply (based on source contributions described in the section "City water supply catchments") (Eq. 1).

$$PPCS_{city} = \sum_{source=1}^{n} PPCS_{source} \times C_{source}; \sum_{source=1}^{n} C_{source} = 1 \qquad (1)$$

Where $PPCS_{city}$ is the Nonpoint PPCS for the city (%), and $PPCS_{source}$ and $C_{source}$ are the PPCS (%) and relative contributions (fraction between 0 and 1) for individual sources supplying the city. This Nonpoint PPCS metric assumes all sources of supply are fully mixed in the public supply system. Groundwater is assumed to be uncontaminated and therefore provides dilution of contaminated water supply, if present. In practice, groundwater may be contaminated. Potential contamination of groundwater sources cannot be readily determined through the geospatial approach followed here and remains a challenge to be addressed in future studies. As demonstrated in Fig. 1, groundwater contributions are marginal for many cities and so results are not highly sensitive to this assumption (although it is of course very important for the few cities that rely heavily on groundwater—see Supplementary Fig. 1 for the impact of groundwater dilution on point and nonpoint PPCS for all cities).

**Flow estimation and computation on effluent proportions at intakes.** Point PPCS is based on the proportion of flow at each intake comprised of effluent from upstream wastewater treatment plants. To determine the effluent volumes discharged to each river system, we sum effluent discharge volumes across all wastewater treatment plants located in the associated watershed upstream of the public supply intake (wastewater treatment plant effluent data are described in the section "Geospatial datasets 450 deployed to assess human activity in watersheds"). These volumes are expressed as a percentage of long-term average flow at the associated intake location. Flow estimates at each intake location can be derived using a river routing model driven by the runoff data described above. Instead, we use flow estimates from the National Hydrography Dataset[31,32]. These are obtained by identifying the USGS reach code associated with each intake and then matching reach codes to the simulated (reanalysis) flow data included in NHDplusV2. We choose these data because they are corrected to USGS gage flows, representing upstream flow regulation and consumption more accurately than flows simulated with a distributed hydrological and river routing model.

Following a previous study[8], we add the WWTP effluent to regulated intake flow before computing potential contaminant volumes (NHDplusV2 simulated flow lacks WWTP discharge contributions). As with Nonpoint PPCS, we determine Point PPCS (%) for each city by taking the average effluent volumes across all supply sources, weighted according to their relative contributions to overall supply and including groundwater contribution (assumed uncontaminated, as with Nonpoint PPCS).

**Addition of hypothetical indirect potable reuse (IPR) scheme.** To illustrate the potential impact of IPR on our PPCS metrics, we select six candidate cities. These include three cities (San Diego, Phoenix, Denver) relying on water withdrawn from the Colorado River Basin, which is currently overallocated, and three fast-growth cities in Texas (Austin, Fort Worth, Houston). San Diego's "Purewater" scheme is currently under construction and serves as a template for how an IPR scheme might be adopted in practice. We copy the following key features of this scheme to explore the possible impacts of IPR on the other cities. First, the potable reuse is indirect, with treated wastewater undergoing further advanced treatment before being returned to the environment where it mixes with naturally available water. The advanced treatment results in a highly purified water source entering supply. Second, an existing supply source is used as the environmental buffer; reclaimed water gets pumped back to a reservoir or upstream of a river intake already used in supply. This is a common approach to IPR when implemented in practice (e.g., "NEWater" in Singapore). Third, as with the San Diego IPR scheme, the reuse schemes implemented in our analysis are designed to meet one third of the city's total water demand. This does not mean that one third of supply will be composed of reclaimed water. Natural inflows to the augmented source reduce the proportion of reclaimed water in the final supply.

We assume reclaimed water has 0% Point PPCS and 0% Nonpoint PPCS. In other words, even though reclaimed water constitutes 100% recycled wastewater, we assign it to an entirely different category of de facto reuse due to the levels of treatment it undergoes before being returned to the environment. However, since consumers may consider this water "potentially contaminated" we also compute a Point PPCS metric that includes de facto reuse and reclaimed water. The use of reclaimed water means that withdrawals from other sources can be reduced, and we assume a uniform reduction of one third from all contributing supply sources. A simplified worked example for a hypothetical two source system is given in Supplemental Materials Fig. 3.

## Data availability

The city-scale potential contamination results[33], water supply contribution data[34], and all geospatial data required to execute this study have been deposited at Zenodo under accession codes 5602059, 4315195, and 4662993, respectively. All of these data repositories are linked in a dedicated meta-repository that supports this manuscript: https://immm-sfa.github.io/Turner-etal_2021_NatureComm/.

## Code availability

As part of the Integrated Multisector, Multiscale Modeling (IM3) project (funded by the U.S. Department of Energy's Office of Science; see "Acknowledgements"), we are committed to delivering all research products as open-source to benefit those who may have interest in reproducing or building off of our work. For the reader's convenience, we provide the open-source code as well as step-by-step documentation for how to reproduce the results and figures represented in this paper in the following meta-repository: https://immm-sfa.github.io/Turner-etal_2021_NatureComm/. The software used to perform this research is known as "gamut"[35] and is available via Github (https://github.com/IMMM-SFA/gamut) and with the appropriate version stored on Zenodo[36].

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

## Acknowledgements

This research was supported by the U.S. Department of Energy, Office of Science, as part of research in MultiSector Dynamics, Earth and Environmental System Modeling Program (Grant No. 59534, supporting S.T., J.R., C.V., K.N., and R.M.). L.M. acknowledges support by the National Science Foundation Grant ACI-1639529 ("INFEWS/T1: Mesoscale Data Fusion to Map and Model the U.S. Food, Energy, and Water (FEW) System"), as well as U.S. Geological Survey Grant/Cooperative Agreement No. G20AP00002 ("Mapping and modeling of interbasin water transfers within the United States") and the U.S. Geological Survey John Wesley Powell Center for Analysis and Synthesis supported working group project ("Reanalyzing and Predicting U.S. Water Use using Economic History and Forecast Data: an experiment in short-range national hydro-economic data synthesis"). We thank Beth Mundy, James Stegen, Cecilia Tortajada (reviewer), and two anonymous reviewers for their valuable feedback and suggestions for improvement. Any opinions, findings, and conclusions, or recommendations expressed in this material are those of the authors and do not necessarily reflect the views of the organizations named above.

## Author contributions

J.R. conceived the study. S.T., J.R., C.V., and R.M. developed the geospatial concept and general approach. L.M. identified the opportunity to focus on de facto reuse and potential contamination of water supply. S.T. developed the PPCS metrics and approach for analyzing impacts of IPR. S.T. and K.N. collected data on surface water source contributions for 116 utilities. S.T., C.V., and K.N. conceptualized and developed the Geospatial Analytics for Multisectoral Urban Teleconnections ("gamut") software used to generate results. S.T. and K.N. generated and analyzed results. S.T. and J.R. led manuscript preparation with contributions from L.M., K.D., C.V., R.M., and K.N.

## Competing interests

The authors declare no competing interests.
