## [Peer Review File · Nature Communications]

Comparison of potential drinking water source contamination across one hundred U.S. citiesReviewers' Comments:

Reviewer #1:

Remarks to the Author:

GENERAL COMMENTS

This paper provides a novel assessment of drinking water source quality for large cities in the conterminous United States. In a creative application of existing data, the authors analyze nonpoint and point source potential contamination in the receiving waters of urban water utilities. They identify a number of different contexts by which urban drinking water may be contaminated using agricultural, industrial, or developed land use data, ultimately finding that 80% of cities intake water that is in part made up of treated wastewater. The authors then discuss the implications this synthesis could have on direct potable reuse adoption.

The paper is well-written and covers a timely and important topic that readers of this journal would find interesting. While I include a few comments below for the authors to consider in this review process, the main consideration I would like the authors to contemplate is whether the work presented here aligns deeply enough with the focus of the discussion around potable reuse (as this is a really interesting bit to consider). In particular, direct potable reuse is made possible in part by having the technology to treat the water to drinking water standards- a real "concentration" based discussion. Yet the paper, understandably so, looks at potential concentration of pollutants via a ratio of affected vs unaffected runoff, not concentrations explicitly. Upon having read this manuscript, I find the jump to making the results presented relevant to the direct potable conversation just a little too far to be very informative. Some suggestions are listed below on how the concepts of actual contamination, potential contamination, and potable reuse might be drawn closer together.

SPECIFIC COMMENTS

Following on the above comment, I was musing on what a few options might be, given the data availability, for bridging that gap. One suggestion would be to include some indication of how often wastewater plants sampled are exceeding standards (i.e., record water quality violations). It could be very helpful to have a sense of how often this is happening upstream. In my mind it would add a layer of nuance to the discussion (i.e., wastewater plants may treat water really well and be less of a concern than other potential sources – which might lend itself to supporting the potable reuse discussion). It might also be (maybe it's an and/or) helpful to have some sense of how often the receiving water utility also experiences violations (i.e., implying that the quality is so bad they can't keep up with it). That would again help readers gauge the vulnerability of these different cities to drinking water contamination. And either of these could potentially add a little more depth to the conversation around our capacity for potable reuse.

While I understand that water quality data isn't explicitly used in this assessment, I find it confusing when the authors refer to the data they produce as a "concentration" of reuse in the water supplied to consumers (e.g., Lines 101-102, 429). It's a ratio of reused water to pristine water- right? Concentration implies a mass per volume of something, yet as far as I can tell this is always a volume to volume ratio?

The authors choose to use a long-term average flow volume in this assessment, but as with any hydrologic process, variability is so important (as the authors note in Lines 167-169). Would it be possible to identify which utilities face greater risks from more variable flows since it would potentially impact the ratio of reused-to-pristine water being withdrawn? I should be fair in saying this addition wouldn't be critical to understanding the overall results, but it would add some clarity to the insights one takes away from this work.

I noticed that the authors used the McDonald 2014 Urban Water Blueprint citation and it reminded me of his 2016 PNAS paper on land use degradation on drinking water source quality:

<https://www.pnas.org/content/early/2016/07/19/1605354113>. It seems relevant to the context of the paper and of interest to the authors.

TECHNICAL COMMENTS

Line 60: To be fair, the same document consumers get reporting water quality for a given utility also by law much include information on where their drinking water comes from (consumer confidence reports). If the authors however meant the origins of the water that make up that drinking water source, vs. the drinking water source location/name, then a little clarity should be added.

Line 100: Perhaps a typo- missing a "the"? "By assessing defacto reuse at THE supply system level..."

Fig 1a (right panel): Is there any reason to have one axis in scientific notation and the other in decimal? Might be easier for the reader if you just pick one format and stick to it.

Fig 1a (left panel): That US map is so TINY! Could it be enlarged even a little this first time it shows up?

Line 377-378: I did not read the cited paper here, but it would be helpful to the reader to know if this is based on US Census data or some other population model.

Reviewer #2:

Most United States cities are exposed to treated wastewater in drinking water supply

Relevant. Analysis of point and non-point potential contamination concentrations in the water supply for 116 large cities

The paper is certainly interesting. It will contribute to the academic literature and also to inform policy. A main problem is the disconnect between the results obtained and the discussion on potable water use. A clear implication of the findings of this study would be, as the authors mention, in terms of exposure of cities (and their population) to different sources of potential water contamination, to and protection efforts. It would also contribute to discussions on health implications, and on regulations and their implementation. For example, the fact that legal limits for contaminants in tap water have not been updated for almost two decades and that the best way to ensure clean tap water is to keep pollution out of source water.

The paper proposes that drinking water supply catchments are increasingly exposed to land use change that causes surface water contamination. While this is true, there are other reasons for surface water contamination including poor management and treatment of point sources of pollution and poor management of non-point sources of pollution.

As mentioned earlier, the paper, including the abstract, does not make a convincing case for direct potable use. In the abstract, the last paragraph says “Four-fifths of cities that withdraw surface water are supplying water that includes a portion of treated wastewater – suggesting that an informed public may consent to direct potable reuse”. One could argue that water utilities in cities identified would first have to analyse whether direct potable reuse is necessary and, if so, if it would be cost-effective. Should this be the case, they would have to start a process that would include public consultation where the population would express their views. Another question would be why direct potable reuse and not indirect potable reuse? What would be the pluses and minuses?

Introduction.

9-45 million people affected is too wide a range.

Not all the contaminants mentioned at the end of the first paragraph are the result of human land use (why human?), such as pharmaceuticals and endocrine disruptors.

The authors mention that there is no comprehensive national assessment of potential contamination of water supplied to large U.S. cities. How does this study compare to the EPA National Aquatic Resource Survey?

“Rather than relying on observed water quality data.” Shouldn’t water quality complement this study that is based on land use? What are the limitations of the findings of this study in the absence of water quality data? How accurate is the evaluation of potential contamination of each city’s total water supply?

A reference is needed for “then blend water from intakes located across multiple local and remoter watersheds and aquifers”.

“Addressing it may soften consumer opposition to water supply alternatives such as direct potable reuse”. Addressing it will be of great use to have a more educated public that could contribute in terms of water conservation, for example, but not to support potable water reuse.

Results. The authors can include names of cities throughout the text. Not all 116 cities studied seem to be represented in the figures.

That less than half of cities supply water studied is generated in relatively pristine land will be welcome. In terms of informing policies, how can this information be used?

“High nonpoint PPCS cases in the South Atlantic tend to arise due to significant urbanization of the watershed—common when drinking water intakes are located close to consumers within city boundaries.” Does this mean that the results would be different if the drinking water intakes were not close to consumers in highly urbanized cities? This is, is the priority the location of the drinking water intakes rather than how urbanized cities are?

About Dallas with the second highest point PPCS due to significant residential sprawl and associated wastewater infrastructure. What is meant by associated wastewater infrastructure? Does it vary from one city to another and how is this important in the context of this study?

The paper rightly refers to the importance of natural diluting river flow. Does this mean that if the study had been carried out any year when there is no snowpack in the state of California, the results would have been different?

In the case of San Francisco, how do the results in the paper compare to EWG drinking water quality reports on contaminants in the San Francisco City Water System?

“Most consumers consider their tap water to be safe”. This statement can be challenged by authors who support that the increasing use of bottled water is because people perceive tap water to be unsafe. Boil water advisories must influence the opinion of the consumer.

For the contributions from groundwater, are these limited to the work by Nelson et al. 2021, or did the authors check the publicly available utility and website information? If this was the case, the authors can include a list of sources and links to the material used that can be made available as supplementary material.

The authors use data sources from different years for their analysis. How does this affect the results obtained and their interpretation? What are the limitations of the several data sources used?

Reviewer #3:

Remarks to the Author:

The authors evaluate point and nonpoint source contributions to surface water sources serving 116 large cities (municipalities serving greater than 150,000 people) across the U.S. In doing so, the authors bring together multiple large datasets to comprehensively examine multiple sources of potential contamination within each delineated watershed. Outcomes of the study expand upon recent studies on de facto reuse (indirect reuse of treated wastewater) to provide insights on the relative contributions of livestock, mining and oil/gas extraction non-point sources within the watershed. The development and application of two metrics (point and nonpoint Proportion of Potentially Contaminated Supply) is a novel contribution that will likely be of interest to others in the field, particularly due to the scale in which the analysis is performed.

There do not seem to be any major flaws in the approach, however it's unclear how the methodology used to estimate runoff in Section 4.3 compares to the EROM used to estimate streamflow within NHDPlus V2 (which is used in Section 4.4). Overall, the manuscript can be improved by the authors being more forthcoming about the limitations of their study, assumptions made, and implications of those assumptions. Some of the model limitations are spread within the methods section, but a separate section further detailing the limitations is warranted. The study aims to evaluate potential exposures to contaminants, but there is no discussion of contaminant concentrations or how those concentrations vary between nonpoint and point sources. Results are presented as the % of the total water supply, but there should be some discussion regarding the expected differences in contaminant concentrations from the various sources (especially when comparing treated ww and nonpoint sources). Similarly, there is a lack of discussion regarding the fate and transport of the contaminants of interest and the underlying assumption of zero instream loss.

More detailed comments are included below:

- (Line 1) The title emphasizes de facto reuse, however this conclusion has already been made in the cited literature on de facto reuse. Instead, I suggest highlighting the more novel outcomes of the study relating to the furthered understanding of nonpoint sources within the title. The authors should also be careful to only draw conclusions within the bounds of the study. For example, 'most cities' is referenced in the title, however only 116 'large' cities are included within the study. This is important because prior work has demonstrated that larger cities are more likely to pull from water sources with DFR, therefore this dataset is somewhat skewed.
- (Line 22) More research is needed to assess the link between DFR occurrence and DPR acceptance before this assertion can be made.
- (Line 75) All contributing sources are not taken into account in this study, in particular this study is limited in regard to point discharges.
- (Line 88-90) This is somewhat misleading because industrial discharges are not accounted for in this study. I suggest changing residential wastewater to municipal wastewater.
- (Lines 186-187) More detail is needed to justify the assertion that the issue is largely avoided.
- (Lines 235-237) Given how far upstream this watershed reaches, are these chemicals expected to be present at the intake?
- (Lines 354-356) How are they found to be accurate? And what level of accuracy?
- (Lines 420-424) This assumption is more limiting for certain contaminants that are known to be present in groundwater, PFAS, for example. Discussing this would be beneficial.
- (Line 442) Does effluent concentration refer to the percentage of effluent to total water supply? The term effluent concentration commonly refers to a specific contaminant and may cause confusion for the reader.

Reviewer #1

Reviewer comment: *This paper provides a novel assessment of drinking water source quality for large cities in the conterminous United States. In a creative application of existing data, the authors analyze nonpoint and point source potential contamination in the receiving waters of urban water utilities. They identify a number of different contexts by which urban drinking water may be contaminated using agricultural, industrial, or developed land use data, ultimately finding that 80% of cities intake water that is in part made up of treated wastewater. The authors then discuss the implications this synthesis could have on direct potable reuse adoption.*

Authors' response: Thank you. We greatly appreciate your detailed review and helpful suggestions for improvement. As you will see from the following point-by-point response, we agree with your suggestions and have taken various steps to improve the paper, including a new analysis of the impact of *indirect potable reuse* on the main metrics developed in our study.

To ensure our interpretation of your comments is correct and our response is clear, we will refer to water reuse in terms of either de facto reuse, indirect potable reuse, or direct potable reuse, as defined in our response to the editor on pages 1 and 2 of this document.

Reviewer comment: *The paper is well-written and covers a timely and important topic that readers of this journal would find interesting. While I include a few comments below for the authors to consider in this review process, the main consideration I would like the authors to contemplate is whether the work presented here aligns deeply enough with the focus of the discussion around potable reuse (as this is a really interesting bit to consider). In particular, direct potable reuse is made possible in part by having the technology to treat the water to drinking water standards- a real "concentration" based discussion. Yet the paper, understandably so, looks at potential concentration of pollutants via a ratio of affected vs unaffected runoff, not concentrations explicitly. Upon having read this manuscript, I find the jump to making the results presented relevant to the direct potable conversation just a little too far to be very informative. Some suggestions are listed below on how the concepts of actual contamination, potential contamination, and potable reuse might be drawn closer together.*

Authors' response: We agree that our discussion on the prospects for potable reuse given consumer awareness of de facto reuse was over-assertive given the state of existing research. We based this discussion on some consumer survey research in Rice et al. (2016) showing a correlation between consumer knowledge of existing de facto reuse and willingness to accept direct or indirect potable reuse. We have since examined existing potable reuse facilities across the U.S., and we find water scarcity to be a stronger driver of direct or indirect potable reuse deployment than water quality.

Actions taken: Since we agree that implications for potable reuse constitute some of the most intriguing aspects of the study, we have added a new analysis to strengthen the link between

potential contamination metrics and potable reuse implementation. This new analysis examines the possible impacts of indirect potable reuse on point and nonpoint PPCS metrics for a selection of cities. We select three cities relying on Colorado River water (San Diego, Denver, Phoenix) and three major cities located in Texas experiencing rapid growth (Austin, Fort Worth, Houston). These are deemed likely candidate cities for indirect potable reuse given the water supply challenges they face. We then implement for each city a hypothetical indirect potable reuse (IPR) scheme which follows the key features of an in-construction scheme in San Diego, namely: IPR to contribute one third of supply; reclaimed water generated with advanced wastewater treatment; treated wastewater pumped to the nearest currently used reservoir (or upstream of the nearest river intake) serving the city, where it is combined with existing water in the reservoir before being drawn into the existing treatment and distribution system (all new calculations detailed in Method).

Although this analysis relies on hypothetical future supply systems, it leads to a number of interesting findings relating to the impacts of IPR on water quality. For example, we show that IPR adoption in Houston would lead to only a very marginal increase in total treated wastewater in supply, because (a) it would reduce reliance on Lake Livingston (high proportion of de facto re use), and (b) the IPR reclaimed water would be diluted significantly when returned to the environment before re-entering the supply system. In a real-world setting, this form of analysis could also be conducted with use of actual contaminant concentrations in place of PPCS metrics—an avenue for future research that lies beyond the scope of the present study. We hope this new analysis closes the gap you identified between our analysis of de facto and potable reuse. We believe the approach we have devised could be used in future, national scale work investigating costs and benefits of indirect or direct potable re-use for urban water supply.

Reviewer comment: *Following on the above comment, I was musing on what a few options might be, given the data availability, for bridging that gap. One suggestion would be to include some indication of how often wastewater plants sampled are exceeding standards (i.e., record water quality violations). It could be very helpful to have a sense of how often this is happening upstream. In my mind it would add a layer of nuance to the discussion (i.e., wastewater plants may treat water really well and be less of a concern than other potential sources – which might lend itself to supporting the potable reuse discussion). It might also be (maybe it's an and/or) helpful to have some sense of how often the receiving water utility also experiences violations (i.e., implying that the quality is so bad they can't keep up with it). That would again help readers gauge the vulnerability of these different cities to drinking water contamination. And either of these could potentially add a little more depth to the conversation around our capacity for potable reuse.*

Authors' response: Thank you for these suggestions. We hope that the new analysis described above helps bridge the gap. We also like the additional analyses you suggest. We have therefore obtained data describing Clean Water Act effluent violations covering all watersheds in this study *and* water supply violations for all of the cities considered in this study.

Actions taken: We have added a new section to results (2.5 “Potential to infer source drinking water contamination risk from PPCS metrics”) which draws on analyses of recorded effluent violations and water supply violations data to highlight limitations of our approach for inferring actual water quality risks for utilities. [LINES 371 – 431]

Reviewer comment: *While I understand that water quality data isn't explicitly used in this assessment, I find it confusing when the authors refer to the data they produce as a "concentration" of reuse in the water supplied to consumers (e.g., Lines 101-102, 429). It's a ratio of reused water to pristine water- right? Concentration implies a mass per volume of something, yet as far as I can tell this is always a volume to volume ratio?*

Authors' response: We agree that the word “concentration” evokes mass per volume concentration of a contaminant rather than volume per volume proportion of water made up by wastewater or non-pristine runoff. In the name of the metric we use “proportion” (i.e., Proportion of Potentially Contaminated Supply—PPCS), so this is the term we now revert to throughout.

Actions taken: We have replaced the word “concentration” with “proportion” throughout the text.

Reviewer comment: *The authors choose to use a long-term average flow volume in this assessment, but as with any hydrologic process, variability is so important (as the authors note in Lines 167-169). Would it be possible to identify which utilities face greater risks from more variable flows since it would potentially impact the ratio of reused-to-pristine water being withdrawn? I should be fair in saying this addition wouldn't be critical to understanding the overall results, but it would add some clarity to the insights one takes away from this work.*

Authors' response: Thank you for this suggestion. Note that the impacts of low flows on de facto reuse and associated pollutants has been analyzed previously for individual intakes:

Rice, J. and Westerhoff, P., 2017. High levels of endocrine pollutants in US streams during low flow due to insufficient wastewater dilution. *Nature Geoscience*, 10(8), pp.587-591.

In a similar fashion to the above study, we are also able to recompute the PPCS using low flows. However there are some major caveats. One issue is reservoirs. Reservoirs would smooth temporary fluctuations in PPCS. The most extreme case is cities using water from the Great Lakes; a period of very low flow and thus high proportion of wastewater in the inflows to, say, Lake Michigan, would have negligible impact on overall proportions of contaminated supply at the water supply intakes immediately downstream on the Detroit River. This effect would occur at varying degrees across supply systems and would be difficult to address without reservoir models including mixing rates and retention times. A second problem is that systems with access to multiple resources may vary the contributions made from each source during drought. For example, groundwater may contribute more to a city's supply during drought. At present we lack data describing source contributions during dry conditions. A final issue is that we lack data on how outflows from wastewater treatment plants change during

drought. The effluent data applied in our study are long-term averages only. Plants dealing with combined sewage and urban surface runoff would likely have significantly reduced outflows during drought, which we are unable to capture.

We can, however, follow your suggestion of identifying which cities might be most vulnerable to increased contamination during low flow periods. These would be cities that rely primarily on direct river withdrawals and which have limited diversity of supply.

Actions taken: We have added a new paragraph to indicate which cities would likely be most vulnerable to PPCS spikes during drought. [LINES 199 – 209]

Reviewer comment: *I noticed that the authors used the McDonald 2014 Urban Water Blueprint citation and it reminded me of his 2016 PNAS paper on land use degradation on drinking water source quality: <https://www.pnas.org/content/early/2016/07/19/1605354113>. It seems relevant to the context of the paper and of interest to the authors.*

Authors' response: Thank you. We are indeed familiar with this work and we agree that it should be included in our paper.

Actions taken: We have added this study to our introduction to support our statement that anthropogenic activity in a watershed increases the costs of treatment and thus the price of water. [LINE 46]

Reviewer comment: *Line 60: To be fair, the same document consumers get reporting water quality for a given utility also by law much include information on where their drinking water comes from (consumer confidence reports). If the authors however meant the origins of the water that make up that drinking water source, vs. the drinking water source location/name, then a little clarity should be added.*

Authors' response: Indeed, the purpose of this passage was to highlight that consumers are generally unaware of the characteristics and prevalence of human activity in the lands upstream of intakes.

Actions taken: We have amended this passage to highlight lack of information on human interactions with surface water upstream of reservoirs and intakes/ [LINES 53 – 56]

Reviewer comment: *Line 100: Perhaps a typo- missing a "the"? "By assessing defacto reuse at THE supply system level..."*

Actions taken: Corrected.

Reviewer comment: *Fig 1a (right panel): Is there any reason to have one axis in scientific notation and the other in decimal? Might be easier for the reader if you just pick one format*

and stick to it. Fig 1a (left panel): That US map is so TINY! Could it be enlarged even a little this first time it shows up?

Actions taken: Thank you. We have enlarged the US map legend and set all axes to decimal notation.

Reviewer comment: *Line 377-378: I did not read the cited paper here, but it would be helpful to the reader to know if this is based on US Census data or some other population model.*

Actions taken: We have amended the text to note that these data are derived from 2010 US Census. [LINE 477]

Reviewer #2

Reviewer comment: *Analysis of point and non-point potential contamination concentrations in the water supply for 116 large cities. The paper is certainly interesting. It will contribute to the academic literature and also to inform policy. A main problem is the disconnect between the results obtained and the discussion on potable water use. A clear implication of the findings of this study would be, as the authors mention, in terms of exposure of cities (and their population) to different sources of potential water contamination, to and protection efforts. It would also contribute to discussions on health implications, and on regulations and their implementation. For example, the fact that legal limits for contaminants in tap water have not been updated for almost two decades and that the best way to ensure clean tap water is to keep pollution out of source water.*

Authors' response: Thank you for your very detailed and constructive review, and for highlighting the implications of our work for source protection. We agree with you that there was a disconnect between results obtained and the discussion on potable water reuse. Since potable reuse is an area of particular interest for the journal, we have added a new quantitative analysis that shows how indirect potable reuse could affect the metrics computed in our study. Details follow in our point-by-point response.

To ensure our interpretation of your comments is correct and our response is clear, we will refer to water reuse in terms of either de facto reuse, indirect potable reuse, or direct potable reuse, as defined in our response to the editor on pages 1 and 2 of this document. As you correctly identify in your review, our paper was unclear on these distinctions.

Reviewer comment: *The paper proposes that drinking water supply catchments are increasingly exposed to land use change that causes surface water contamination. While this is true, there are other reasons for surface water contamination including poor management and treatment of point sources of pollution and poor management of non-point sources of pollution.*

Authors' response: We agree that it is important to cover the effects of standards of effluent treatment and land management on contamination arising from human activity in watersheds. One way to capture this is through analysis of Clean Water Act data covering effluent violations, which we have now obtained. These data can help us infer the quality of land and facility management across watersheds serving cities. For example, we find that Fort Wayne, Indiana, which has a developed watershed land fraction of ~10% has very high effluent violation intensity in its watersheds. Raleigh, North Carolina, has a similar developed watershed land fraction (~16%), but relatively low violation intensity—an order of magnitude lower than Fort Wayne. Although not conclusive (i.e., reported violations may not accurately reflect actual violations), this analysis highlights the potential impact and importance of contaminant source management.

Actions taken: We have added a new discussion (section 2.5) along with four new figures in supplemental material comparing metrics computed in our study (potential contamination

metrics, land area developed for anthropogenic activity, number of mining facilities) against the intensity of reported effluent violations arising from different sectors (using number of reported violations per km² watershed area) [LINES 371 - 431].

Reviewer comment: As mentioned earlier, the paper, including the abstract, does not make a convincing case for direct potable use. In the abstract, the last paragraph says “Four-fifths of cities that withdraw surface water are supplying water that includes a portion of treated wastewater – suggesting that an informed public may consent to direct potable reuse”. One could argue that water utilities in cities identified would first have to analyse whether direct potable reuse is necessary and, if so, if it would be cost-effective. Should this be the case, they would have to start a process that would include public consultation where the population would express their views. Another question would be why direct potable reuse and not indirect potable reuse? What would be the pluses and minuses?

Authors’ response: Thank you. Our intention with this sentence at the end of the abstract was not to make a convincing case for potable reuse, but instead to highlight the possible impact our results could have on public opinion of potable reuse. To make this inference we drew on previous survey research that suggests consumers are more likely to accept *indirect* or *direct* potable reuse if they are aware of existing *de facto* reuse in their supply. We agree that this inference is too weak and that more can be done to place our findings in the context of direct and indirect potable reuse. We also agree indirect reuse should be addressed, since this form of potable reuse appears to be more common in real-world applications of reuse—including the in-construction scheme for San Diego as well as other globally important cases, such as Singapore.

Actions taken: While we do not wish to take any advocacy position on direct or indirect potable reuse, we can use our results to explore possible impacts of such schemes on the metrics developed in our study. To do this, we have added a new analysis that examines the possible impacts of indirect potable reuse (IPR) on point and nonpoint PPCS metrics for a selection of cities. We select three cities relying on Colorado River water (San Diego, Denver, Phoenix) and three major cities located in Texas experiencing rapid growth (Austin, Fort Worth, Houston). These are deemed likely candidate cities for indirect potable reuse given the water supply challenges they face. We then model for each city a hypothetical indirect potable reuse scheme which follows the key features of an in-construction scheme in San Diego, namely: IPR to contribute one third of supply; reclaimed water generated with advanced wastewater treatment; treated wastewater pumped to the nearest currently used reservoir (or upstream of the nearest river intake) serving the city, where it is combined with existing water in the environment before being drawn into the existing treatment and distribution system (all new calculations detailed in Method). This new analysis allows us to explore the impacts of an IPR scheme on PPCS metrics across a range of cities and also highlight some of the design considerations that would be faced by utilities implementing such schemes. [LINES 317 – 370]

Reviewer comment: 9-45 million people affected is too wide a range.

Authors' response: This range is depending on year and was taken directly from the cited study.

Actions taken: We have changed this to read “up to 45 million people.” [LINE 32]

Reviewer comment: Not all the contaminants mentioned at the end of the first paragraph are the result of human land use (why human?), such as pharmaceuticals and endocrine disruptors.

Authors' response: Thank you. We were considering point effluent sites also part of “land use”. We agree this should be clarified.

Actions taken: Changed to simply “anthropogenic activity.” [LINE 44]

Reviewer comment: The authors mention that there is no comprehensive national assessment of potential contamination of water supplied to large U.S. cities. How does this study compare to the EPA National Aquatic Resource Survey?

Authors' response: The EPA NARS provides information for a selection of lakes and streams spread throughout CONUS. This includes chemical contaminant concentrations relevant to our study. NARS aims to capture a statistically significant sample for lakes and streams covering the entire United States; the survey is not designed to capture drinking water specifically, nor watersheds supplying individual large cities specifically.

Actions taken: We have provided further detail in our introduction to differentiate our approach from existing water quality surveys, including NARS and the 303d impaired water body list. [LINES 49 – 60]

Reviewer comment: “Rather than relying on observed water quality data.” Shouldn't water quality complement this study that is based on land use? What are the limitations of the findings of this study in the absence of water quality data? How accurate is the evaluation of potential contamination of each city's total water supply?

Authors' response: There are certainly limitations in our approach for inferring actual contamination, and we agree that these should be drawn out and emphasized. One limitation is related to your above comment on differences in quality of management of land and possible polluting resources across watersheds and cities. This is one we can begin to address through analysis of effluent violations. Another key limitation, raised by reviewer #3, is lack of accounting for natural removal of contaminants in very large watersheds. Section 2.5 in the revised manuscript is dedicated to highlighting and discussing these limitations.

As to whether observed water quality could be introduced, we have the following thoughts. First, it *would* be possible to develop measured water quality data for a selection of intakes and reservoirs (though not all). Having examined the relevant datasets (USGS Water Quality Portal combines multiple records) we believe this would constitute a substantial workload. Large numbers of candidate stations near intakes, inconsistent labeling of measured

parameters, large number of parameters, varying periods of record, etc., make these data difficult to work with at the scale of this study—these are more appropriate for watershed / county / state scale analyses. Prior national scale applications of these data focus on just one or two parameters (e.g., N and P) and do not offer the comprehensive suite of potential contaminants that we intend to represent through our analysis of anthropogenic activity. In addition, water quality surveys tend to focus on regulated or well recognized contaminants, potentially missing important contaminants.

Our study focuses primarily on human surface water interactions in water supply catchments—a detail that is missing in publicly available Consumer Confidence Reports—and how large cities across the U.S. differ in their degree and source of exposure to potential contamination. Importantly, measured contaminant concentrations would not alter the PPCS metrics evaluated in our study. These data could inform further analysis of limitations, but we believe the use of violations data and discussion of limitations relating to physical and institutional factors suffices in informing the reader that potential contamination is not necessarily commensurate with actual. We do see great potential for a substantial future study informed by measured contamination (e.g., a modeling study using PPCS and other data to predict measured concentrations) as well as further development of the IPR analysis with measured contaminant concentrations replacing PPCS metrics for utility scale studies.

Actions taken: We have added a new section covering the limitations of the approach for inferring actual water quality and contamination risk, offering analysis and discussion on the likely disconnect between PPCS metrics and actual contamination, based partly on analysis of effluent and water supply violations for each city [LINES 371 – 431]. We have also altered the introduction to motivate our approach by pointing out the limitations of water quality surveys, which are piecemeal and limited in their coverage—often focusing on regulated contaminants only. [LINES 55 – 62]

Reviewer comment: A reference is needed for “then blend water from intakes located across multiple local and remoter watersheds and aquifers”.

Actions taken: Citation added. [LINE 74]

Reviewer comment: “Addressing it may soften consumer opposition to water supply alternatives such as direct potable reuse”. Addressing it will be of great use to have a more educated public that could contribute in terms of water conservation, for example, but not to support potable water reuse.

Authors’ response: To make this claim that public education on existing water could soften opposition to water reuse, we relied on a prior study showing a correlation between knowledge that . In simple terms: if you learn you’re already consuming significant de facto reuse, why object to planned potable reuse (direct or indirect)? On reflection, we agree that existing consumer research is insufficient to make this claim.

Actions taken: We have removed statements linking public knowledge of existing potential contamination and potable reuse acceptance.

Reviewer comment: Results. The authors can include names of cities throughout the text. Not all 116 cities studied seem to be represented in the figures.

Authors' response: Our supplementary figure shows point and non PPCS metrics for all cities studied (with cities categorized by census division).

Actions taken: We have added a reference to the SI figure for readers interested in the full suite of results. [*LINE 71*]

Reviewer comment: That less than half of cities supply water studied is generated in relatively pristine land will be welcome. In terms of informing policies, how can this information be used?

Authors' response: This result suggests that many cities can avoid future water quality problems by continuing to protect and manage human activity in their source watersheds. We have some tentative plans to combine these watersheds with spatial land-use land cover projections to explore possible issues arising in future as population growth and sprawl encroach into water supply catchments.

Actions taken: We have added a sentence to highlight a potential future issue in Western cities as growth places new pressures to use these lands. [*LINES 253 – 256*]

Reviewer comment: “High nonpoint PPCS cases in the South Atlantic tend to arise due to significant urbanization of the watershed—common when drinking water intakes are located close to consumers within city boundaries.” Does this mean that the results would be different if the drinking water intakes were not close to consumers in highly urbanized cities? This is, is the priority the location of the drinking water intakes rather than how urbanized cities are?

Authors' response: Results would certainly be different. Atlanta is a good example. Water availability is regulated from a reservoir located north of the city. But the drinking water intake is downstream, within city boundaries. Water supply is therefore a combination of reservoir release and local urban runoff. In this case, and others like it (Dallas, Texas), the city has expanded into the watershed over a number of years and decades. There is inertia in the infrastructure; the intake is located next to an existing water treatment facility. Moving the intake upstream would require a new water treatment facility or a major transfer project to deliver that water to the existing treatment plant.

Note that one could apply our model to assess how PPCS metrics change with intakes moved further upstream or onto neighboring river systems.

Actions taken: We have added discussion on the potential of our approach to explore alternative intake locations and associated impacts on potential contamination. [LINES 319 – 321]

Reviewer comment: About Dallas with the second highest point PPCS due to significant residential sprawl and associated wastewater infrastructure. What is meant by associated wastewater infrastructure? Does it vary from one city to another and how is this important in the context of this study?

Authors' response: Thank you—this was sloppy use of the term wastewater infrastructure; we simply meant that wastewater treatment plants are built in these new outcrops of the city, leading to more de facto reuse in rivers, ultimately ending up back in supply.

Actions taken: We have changed “wastewater infrastructure” to “wastewater treatment facilities plants discharging within Dallas’s water supply catchments.” [LINES 182 – 183]

Reviewer comment: The paper rightly refers to the importance of natural diluting river flow. Does this mean that if the study had been carried out any year when there is no snowpack in the state of California, the results would have been different?

Authors' response: Yes. Our study is based on long-term averages not any individual year. Results would certainly be different in dry years. Although we have the capability to update our analysis with dry year runoff and flows, we are unable provide reasonable estimates of metrics for given drought years. One problem is reservoirs, which would smooth temporary fluctuations in PPCS. The most extreme cases are cities using water from the Great Lakes; a period of very low flow and thus high proportion of wastewater in the inflows to, say, Lake Michigan, would have negligible impact on overall proportions of contaminated supply at the water supply intakes immediately downstream on the Detroit River. This effect would occur at varying degrees across supply systems and would be difficult to address without reservoir models including mixing rates and retention times. A second problem is that systems with access to multiple resources may vary the contributions made from each source during drought. For example, groundwater may contribute more to a city’s supply during drought. At present we lack data describing source contributions during dry conditions. A final issue is that we lack data on how outflows from WWTPs change during drought. The effluent data applied in our study are long-term averages only. Plants dealing with combined sewage and urban surface runoff would likely have significantly reduced outflows during drought, which we are unable to capture.

Actions taken: We have provided new discussion on which types of cities would be most vulnerable to spikes in PPCS during dry years, highlighting cases where supply is withdrawn directly from a river and for which no alternative resources are available for supply. [LINES 199 – 209]

Reviewer comment: In the case of San Francisco, how do the results in the paper compare to EWG drinking water quality reports on contaminant in the San Francisco City Water System?

Authors' response: For the San Francisco Water System, EWG's results are consistent with our study. EWG reports three contaminants reported above EWG's limits (though below legal limits). These are:

- Chromium (hexavalent). This contaminant can occur naturally and does not necessarily indicate human-caused pollution.
- Haloacetic acids (HAA5). This contaminant relates to chlorination (occurring post-treatment, and not part of our analysis).
- Total trihalomethanes. As above—relates to the disinfection process, post-treatment.

Actions taken: In a related amendment in response to Reviewer #1's suggestion, we have included new analysis of EPA health-based violations, revealing an interesting link between nonpoint PPCS and occurrence of violations. [LINES 402 – 418]

Reviewer comment: “Most consumers consider their tap water to be safe”. This statement can be challenged by authors who support that the increasing use of bottled water is because people perceive tap water to be unsafe. Boil water advisories must influence the opinion of the consumer.

Authors' response: We agree that there are a range of viewpoints on the quality of tap water and that, as you point out, many people consider their tap water unsafe, even in cities with well-run utilities.

Actions taken: We have removed this element of the discussion section, because it relied on too much inference. This has now been replaced with discussion on possible applications of our approach, including a quantitative analysis of how PPCS metrics could change with implementation of indirect potable reuse in a selection of key cities. [LINES 317 – 370]

Reviewer comment: For the contributions from groundwater, are these limited to the work by Nelson et al. 2021, or did the authors check the publicly available utility and website information? If this was the case, the authors can include a list of sources and links to the material used that can be made available as supplementary material.

Authors' response: Nelson et al. (2021) is actually the citation for our own data product, created for this very study (we are citing the actual input data rather than a previous paper). Contributions from groundwater were obtained by searching through utility websites and other sources if necessary. The dataset includes a list of websites covering all 116 cities in a dedicated file named “Watershed_Contributions_Notes.csv”.

Actions taken: We have removed the citation to Nelson et al. (2021), adding our input data to an “Open Resource” section instead. Our text now states that we have “... listed all websites and grey literature sources in a supporting file included in our input data.” [LINES 455 – 456]

Reviewer comment: The authors use data sources from different years for their analysis. How does this affect the results obtained and their interpretation? What are the limitations of the several data sources used?

Authors’ response: We used to most up to date information available for all data sources applied in the study. Many datasets are available only at specific intervals, such as Census-derived data (once per decade), wastewater treatment plant discharges (last survey was 2012), and the cropland data layer. The variables being estimated (land use, infrastructure, facilities, population) tend to change gradually over years and decades rather than abruptly from one year to the next.

Actions taken: We have highlighted the limitation of different data years and noted that the associated variables tend to change gradually—meaning results should be robust. [LINES 124 - 128]

Reviewer #3

Reviewer comment: The authors evaluate point and nonpoint source contributions to surface water sources serving 116 large cities (municipalities serving greater than 150,000 people) across the U.S. In doing so, the authors bring together multiple large datasets to comprehensively examine multiple sources of potential contamination within each delineated watershed. Outcomes of the study expand upon recent studies on de facto reuse (indirect reuse of treated wastewater) to provide insights on the relative contributions of livestock, mining and oil/gas extraction non-point sources within the watershed. The development and application of two metrics (point and nonpoint Proportion of Potentially Contaminated Supply) is a novel contribution that will likely be of interest to others in the field, particularly due to the scale in which the analysis is performed.

Authors' response: Thank you for your constructive review and helpful suggestions. We appreciate and agree with all of the points you raise below, although there is one minor misinterpretation we wish to correct (see point-by-point response, below).

Reviewer comment: There do not seem to be any major flaws in the approach, however it's unclear how the methodology used to estimate runoff in Section 4.3 compares to the EROM used to estimate streamflow within NHDPlus V2 (which is used in Section 4.4).

Authors' response: Runoff and flow are two entirely separate data products in this analysis, although both are representing average water conditions for a similar, multi-decadal period in history. The runoff data are gridded, high-resolution estimates of land surface runoff only. In other words, this water is not routed to streamflow. We use the runoff data solely to determine the overall proportion of water being generated over the land surface occurring on different types of land (cropland, urban, pristine).

To compute flows at intakes and into reservoirs, we could have used the same runoff data to drive a river routing model, leading to simulated flows consistent with runoff. Instead we use EROM. The reasons for using EROM are: (1) these data are adjusted to USGS gages, thus representing water consumption and regulation that would affect flows at intakes and into reservoirs; (2) EROM are simulated for specific stream locations at national scale, defined by NHD flowlines that are easily linked to intake and reservoir locations (this is superior to our simulation models for converting runoff to flow, which lead to a gridded representation of flow). Importantly, nonpoint PPCS uses the runoff data and point PPCS uses the EROM flows. Neither metric uses both datasets. Since these two metrics are computed in isolation and are not combined in our analytics, the use of two separate water availability products does not pose any consistency problem in our results or analysis.

Actions taken: We have added a passage to Method to justify use of EROM and to highlight the distinction between the two separate data products adopted. [LINES 541 - 543]

Reviewer comment: Overall, the manuscript can be improved by the authors being more forthcoming about the limitations of their study, assumptions made, and implications of those

assumptions. Some of the model limitations are spread within the methods section, but a separate section further detailing the limitations is warranted. The study aims to evaluate potential exposures to contaminants, but there is no discussion of contaminant concentrations or how those concentrations vary between nonpoint and point sources. Results are presented as the % of the total water supply, but there should be some discussion regarding the expected differences in contaminant concentrations from the various sources (especially when comparing treated ww and nonpoint sources).

Authors' response: We agree that expected discrepancies between our metrics based on presence of anthropogenic activity and actual contamination should be elaborated in detail.

Actions taken: We have added an entire section (2.5) devoted to limitations of PPCS metrics for inferring actual contamination and discussion of how PPCS should relate to actual contamination. To support this discussion, we examine reported water supply violations and effluent violations occurring across all watersheds in this study. We also describe various physical characteristics that would affect actual contamination. [LINES 371 – 431]

Reviewer comment: Similarly, there is a lack of discussion regarding the fate and transport of the contaminants of interest and the underlying assumption of zero instream loss.

Authors' response: We agree that this limitation must be highlighted.

Actions taken: Within the new section (2.5), we have included discussion on the review work of Lintern et al. (2018) [<https://doi.org/10.1002/wat2.1260>] to inform the reader on various complexities relating to contaminant mobilization and delivery which could lead to a disconnect between PPCS metrics and actual contamination. [LINES 375 – 381]

Reviewer comment: (Line 1) The title emphasizes de facto reuse, however this conclusion has already been made in the cited literature on de facto reuse. Instead, I suggest highlighting the more novel outcomes of the study relating to the furthered understanding of nonpoint sources within the title. The authors should also be careful to only draw conclusions within the bounds of the study. For example, 'most cities' is referenced in the title, however only 116 'large' cities are included within the study. This is important because prior work has demonstrated that larger cities are more likely to pull from water sources with DFR, therefore this dataset is somewhat skewed.

Authors' response: We agree that the title should reflect both the point and nonpoint aspects of the study. We also agree that key highlights should be worded to reflect our study focuses on large cities.

Actions taken: We have changed the title to “Anthropogenic activities in source watersheds lead to wide disparities in potential contamination of urban drinking water supplies in the United States”

Reviewer comment: (Line 22) More research is needed to assess the link between DFR occurrence and DPR acceptance before this assertion can be made.

Authors' response: We agree that the possible link between DFR occurrence and DPR acceptance was overstated in our first submission.

Actions taken: We have removed discussion on the link between DFR and DPR. In its place, we offer a new quantitative analysis of hypothetical Indirect Potable Reuse (IPR) for a selection of water-stressed cities. [LINES 317 – 370]

Reviewer comment: (Line 75) All contributing sources are not taken into account in this study, in particular this study is limited in regard to point discharges.

Authors' response: Our use of the word “sources” in this paper to refer to both *sources of contamination* and *sources of water* (i.e., different watersheds serving each city) has led to an understandable misinterpretation here. When we say “all contributing sources” we mean that this study is considering all water supply catchments supplying each individual city.

Actions taken: We have reworded to “In contrast to previous research focused on individual water treatment plant intake locations (e.g., Rice and Westerhoff, 2015), we evaluate the potential contamination of each city’s total water supply, accounting for the relative contributions of water supply sources from multiple watersheds.” [LINES 81 – 82]

We have also added an equation to method to highlight how each city’s PPCS is computed from individual watershed PPCS and water source contributions. [LINES 518 – 521]

Reviewer comment: (Line 88-90) This is somewhat misleading because industrial discharges are not accounted for in this study. I suggest changing residential wastewater to municipal wastewater.

Actions taken: Changed “residential” to “municipal”. [LINE 100]

Reviewer comment: (Lines 186-187) More detail is needed to justify the assertion that the issue is largely avoided.

Authors' response: Thank you—agreed. We have looked into max nitrate violations for all cities included in our study. We found that the cities of Fresno and Modesto *do* have recorded violations. But these cities rely on local groundwater (not evaluated in our study). The CA cities relying solely on surface water transferred from relatively pristine basins have no violations on record.

Actions taken: We have rewritten this section to reflect our analysis of recorded nitrate violations. [LINES 223 – 231]

Reviewer comment: (Lines 235-237) Given how far upstream this watershed reaches, are these chemicals expected to be present at the intake?

Authors' response: There seems to be a research gap on fate of shale gas extraction contaminants. From a recent review article focused on shale gas impacts on water quality:

*“the impact of organic compounds from shale gas HF wastewaters on drinking water sources deserves special attention. **There are limited studies on the human toxicity thresholds, fate of organic contaminants in surface waters, and their dilution/attenuation before water intake for drinking water treatment.** More detailed studies on human toxicity assessment of chemicals in HF wastewaters and potential transformation byproducts from wastewater-originated precursors in water sources are needed...HF wastewater contains not only initially added fracturing additives and geologically originated compounds, but also intended and unintended transformation intermediates/by-products generated during the HF process.”* – Sun et al., 2019 (<https://doi.org/10.1016/j.envint.2019.02.019>)

Actions taken: We have added discussion on research gaps relating to shale gas contaminants as well as contaminant fate generally. [LINES 421 – 429]

Reviewer comment: (Lines 354-356) How are they found to be accurate? And what level of accuracy?

Authors' response: These lines state that “we find UWB groundwater contribution estimates to be *inaccurate*”, not *accurate*. We checked utility websites and planning documents for all 116 cities to (a) ensure that the Urban Water Blueprint correctly represented the resources supplying each city (which they do), and (b) check the surface / groundwater split stated in UWB. We found those surface / groundwater splits to be arbitrary in many cases (e.g., if groundwater is used, it's assumed to supply 50% of water). Our investigation of source contributions updates these arbitrary estimates with actual reported use of each surface water resources and any groundwater.

Actions taken: We have added a short note to highlight the arbitrary nature of UWB groundwater / surface water split. [LINES 453 – 454]

Reviewer comment: (Lines 420-424) This assumption is more limiting for certain contaminants that are known to be present in groundwater, PFAS, for example. Discussing this would be beneficial.

Authors' response: We agree completely. This is also a limiting assumption for groundwater dependent regions in lands heavily exploited for crop production. Our new analysis of water supply violations finds two categories of city with recorded max nitrate violations: (1) those high in our nonpoint PPCS metrics, and (2) cities like Modesto and Fresno (California), which have very low nonpoint PPCS but rely heavily on local groundwater sources, leading to recorded nitrate violations.

Actions taken: We now use water supply violations data to highlight the potential importance of contamination from groundwater, and this is also now highlighted as a key limitation in section 2.5. [LINES 418 - 420]

Reviewer comment: (Line 442) Does effluent concentration refer to the percentage of effluent to total water supply? The term effluent concentration commonly refers to a specific contaminant and may cause confusion for the reader.

Authors' response: Yes, you interpreted this correctly. We agree that the word “concentration” is inappropriate.

Actions taken: We have removed all instances of “concentration” throughout the text. This is replaced by “proportion” of effluent in total water supply (in line with the name of the metrics—“Proportion of Potentially Contaminated Supply”).

Reviewers' Comments:

Reviewer #1:

Remarks to the Author:

I appreciate the thoughtful and thorough responses to the comments in this review. In my opinion, all of my questions and concerns regarding this manuscript have been addressed. In particular:

The clarification of terms (e.g. "concentration" vs "proportion" and types of reuse) has made the study results easier to understand.

I found the new sections on water quality violations and hypothetical IPR particularly interesting, adding depth and context to the importance of the work.

And while I agree with the authors that there would be some major caveats to tackling the "low flow" issue that would push such an investigation outside of the scope of this work, I thought the commentary they added to acknowledge this phenomenon, and provide some insights on how variability in flow might impact PPCSs, was helpful.

Finally, I have only one technical comment to make. The Figure numbers in the narrative seem to be unclear as submitted (e.g., check lines 147, 171, 222, 235, 244, etc. where two figures are referenced at the same time). I wasn't sure if they were both intentionally listed, or if it was a track changes error.

I think this will be an impactful and inspiring study, once published.

Reviewer #2:

Remarks to the Author:

The paper has improved significantly. Arguments that were not robust enough have been either revised or deleted. Discussions do advance understanding on the topic of study.

My only comment refers to the authors' decision not to study any city that actually receives water from IPR schemes (Section 3.5) such as Orange country, El Paso, or any other. This decision should be justified.

Reviewer #3:

Remarks to the Author:

In the resubmitted paper the authors have provided further clarifying details regarding their use of terminology, expanded the analysis to include an exploration into the predictive nature of the PPCS metric and case-study into a hypothetical IPR scheme. As well as, expanded the discussion regarding the limitations of the study on contamination prediction. By doing so, the resulting paper has satisfied my prior comments and improved the quality of the article.

Reviewer #1: *I appreciate the thoughtful and thorough responses to the comments in this review. In my opinion, all of my questions and concerns regarding this manuscript have been addressed. In particular:*

- *The clarification of terms (e.g. "concentration" vs "proportion" and types of reuse) has made the study results easier to understand.*
- *I found the new sections on water quality violations and hypothetical IPR particularly interesting, adding depth and context to the importance of the work.*
- *And while I agree with the authors that there would be some major caveats to tackling the "low flow" issue that would push such an investigation outside of the scope of this work, I thought the commentary they added to acknowledge this phenomenon, and provide some insights on how variability in flow might impact PPCSs, was helpful.*

Finally, I have only one technical comment to make. The Figure numbers in the narrative seem to be unclear as submitted (e.g., check lines 147, 171, 222, 235, 244, etc. where two figures are referenced at the same time). I wasn't sure if they were both intentionally listed, or if it was a track changes error.

I think this will be an impactful and inspiring study, once published.

Authors' response: We have corrected the Figure referencing error in the final submission.

Reviewer #2: *The paper has improved significantly. Arguments that were not robust enough have been either revised or deleted. Discussions do advance understanding on the topic of study. My only comment refers to the authors' decision not to study any city that actually receives water from IPR schemes (Section 3.5) such as Orange country, El Paso, or any other. This decision should be justified.*

Authors' response: Existing reclaimed water schemes are typically either experimental (and thus very small scale) or are not actually *potable* reuse. Both El Paso and Orange County are supplying water for non-potable use only. We have added the following sentence to Methods: *"Existing reclaimed water schemes are not included in these contributions, since such schemes tend to be either very small scale (usually experimental projects) or are providing water for non-potable use only (e.g., landscape irrigation)."*

Reviewer #3: *In the resubmitted paper the authors have provided further clarifying details regarding their use of terminology, expanded the analysis to include an exploration into the predictive nature of the PPCS metric and case-study into a hypothetical IPR scheme. As well as, expanded the discussion regarding the limitations of the study on contamination prediction. By doing so, the resulting paper has satisfied my prior comments and improved the quality of the article.*

Authors' note: No further changes requested.